# LEARNING WITHOUT FORGETTING FOR VISION-LANGUAGE MODELS

## ABSTRACT

Class-Incremental Learning (CIL) or continual learning is a desired capability in the real world, which requires a learning system to adapt to new tasks without forgetting former ones. While traditional CIL methods focus on *visual* information to grasp core features, recent advances in Vision-Language Models (VLM) have shown promising capabilities in learning generalizable representations with the aid of *textual* information. However, when continually trained with new classes, VLMs often suffer from catastrophic forgetting of former knowledge. Applying VLMs to CIL poses two major challenges: **1)** how to adapt the model without forgetting; and **2)** how to make full use of the multi-modal information. To this end, we propose PROjectiOn Fusion (**PROOF**) that enables VLMs to learn without forgetting. To handle the first challenge, we propose training task-specific projections based on the frozen image/text encoders. When facing new tasks, new projections are expanded, and former projections are fixed, alleviating the forgetting of old concepts. For the second challenge, we propose the fusion module to better utilize the cross-modality information. By jointly adjusting visual and textual features, the model can capture better semantic information. Extensive experiments on nine benchmark datasets with various continual learning scenarios (CIL and continual cross-modal retrieval) and various VLMs validate PROOF achieves state-of-the-art performance.

## 1 INTRODUCTION

In our ever-changing world, training data often comes in a stream format with new classes, requiring a learning system to absorb them continually (Gomes et al., 2017; Geng et al., 2020). To address the challenge of learning emerging new classes, Class-Incremental Learning (CIL) has been proposed (Rebuffi et al., 2017). However, in CIL, the absence of former classes triggers catastrophic forgetting (French, 1999), where learning new concepts overwrites the knowledge of old ones and results in a decline in performance (Li & Hoiem, 2016). Numerous efforts have been made (De Lange et al., 2021; Masana et al., 2022) to combat catastrophic forgetting in the machine learning field.

With the rapid development of pre-training techniques (Han et al., 2021), recent years have witnessed the transition of CIL research from training from scratch (Wu et al., 2019; Zhao et al., 2020) to *utilizing pre-trained models* (PTM) (Wang et al., 2022c;d). With the help of PTM, *e.g.*, Vision Transformers (Dosovitskiy et al., 2020), incremental learners are born with strong transferability to grasp the *visual* features. Facing the domain gap introduced by the incremental classes, they only need to learn a limited number of additional parameters (Jia et al., 2022) as the *patches* to bridge the distribution gap, which significantly simplifies the challenge of incremental learning.

While pre-trained ViT-based CIL methods focus on learning the *visual* features to recognize new concepts, recent advances in Vision-Language Models (VLM) have demonstrated the potential of *textual* information in building generalized feature representations. A seminal work, *i.e.*, contrastive language-image pre-training (Radford et al., 2021) (CLIP), maps the visual and textual information in the shared embedding space, enabling robust learning and recognition of concepts from diverse sources. This integration of visual and textual modalities presents a promising avenue for developing continual learning models that can effectively adapt to real-world scenarios.

Extending VLMs to CIL faces two significant challenges. First, sequentially tuning the VLM overwrites the innate generalizability and former concepts, leading to forgetting and poor performance on future tasks. Second, relying solely on textual information for classification neglects the valuable

cross-modal features present in the multi-modal inputs. To fully utilize this information, it is necessary to explore methods for cross-modal fusion beyond textual features.

Correspondingly, we aim to turn a VLM into a continual learner that is both *retentive* and *comprehensive*. Retentive refers to the model's ability to maintain its pre-trained capabilities, thereby preserving generalizability and enabling it to perform well on future tasks without forgetting. Comprehensive refers to the model's capacity to integrate and adjust information from multiple modalities. By leveraging these characteristics, we can mitigate catastrophic forgetting and use cross-modal features to build more robust classifiers as data evolves.

In this paper, we propose PROjectiOn Fusion (**PROOF**) to address catastrophic forgetting in VLM. To make the model retentive, we freeze the pre-trained image/text backbones and append liner projections on top of them. The task-specific information is encoded in the corresponding projection layer by mapping the projected features. When facing new tasks, new projections are extended while old ones are frozen, preserving former knowledge. Besides, we aim to fuse the information from different modalities via cross-model fusion, which allows for the query embedding to be adjusted with context information. Consequently, PROOF efficiently incorporates new classes and meanwhile resists forgetting old ones, achieving state-of-the-art performance on nine benchmark datasets. We also evaluate PROOF in various continual learning settings, including CIL and continual cross-modal retrieval, to show its effectiveness in various real-world scenarios.

## 2 RELATED WORK

**Vision-Language Model (VLM) Tuning:** Recent years have witnessed the prosperity of research in VLMs (Radford et al., 2021; Li et al., 2023; Alayrac et al., 2022). With great generalizability, they can be applied for downstream tasks in a zero-shot manner. However, a domain gap still exists between the pre-trained and downstream datasets, requiring further tuning. CoOp and CoCoOp (Zhou et al., 2022b;a) apply prompt learning (Li & Liang, 2021) into VLM tuning with learnable prompt tokens. Subsequent works explore VLM tuning via adapter tuning (Gao et al., 2021), prompt distribution learning (Lu et al., 2022), similarity learning (Zhang et al., 2022), descriptor learning (Mao et al., 2022), and optimal transport mapping (Chen et al., 2023). However, they focus on adapting VLM to downstream tasks while overlooking the *forgetting* of former ones.

**Class-Incremental Learning (CIL):** aims to learn from evolutive data and absorb new knowledge without forgetting. Replay-based methods (Luo et al., 2023; Aljundi et al., 2019; Chaudhry et al., 2018a; Liu et al., 2020; Chaudhry et al., 2018b) save and replay former instances to recover old knowledge when learning new ones. Knowledge distillation-based methods (Rebuffi et al., 2017; Li & Hoiem, 2016; Douillard et al., 2020) build the mapping between models as regularization. Parameter regularization-based methods (Kirkpatrick et al., 2017; Aljundi et al., 2018; Zenke et al., 2017) weigh the importance of different parameters as regularization. Model rectification-based methods (Shi et al., 2022; Zhao et al., 2020; Wu et al., 2019; Yu et al., 2020) rectify the inductive bias for unbiased predictions. Dynamic networks (Yan et al., 2021; Wang et al., 2022a; Zhou et al., 2023b) show strong performance by expanding the network structure as data evolves. (Deng et al., 2021; Saha et al., 2021; Lin et al., 2022) explore gradient projection in new tasks to enable model updating without harming former knowledge. In contrast to these works, in this paper, we propose to learn feature projections based on the frozen embedding functions to encode task-specific information.

**CIL with VLM:** The aforementioned CIL methods aim to train an incremental model from scratch, while it would be easier to start with a pre-trained model (Lee et al., 2023). The integration of pre-trained Vision Transformer (Dosovitskiy et al., 2020) into CIL has attracted the attention of the community, and most methods (Wang et al., 2022c;d; Seale Smith et al., 2022) employ parameter-efficient tuning techniques to learn without forgetting. S-Prompt (Wang et al., 2022b) explores CLIP in *domain*-incremental learning, but the application of VLM in CIL remains relatively unexplored. WiSE-FT (Wortsman et al., 2022) utilizes weight ensemble for robust finetuning, while it cannot be extended to multiple tasks.

## 3 PRELIMINARIES

### 3.1 CLASS-INCREMENTAL LEARNING AND VISION LANGUAGE MODEL

In CIL, we have the sequence of $B$ training sets without overlapping classes, denoted as $\left\{\mathcal{D}^1, \mathcal{D}^2, \cdots, \mathcal{D}^B\right\}$, where $\mathcal{D}^b = \{(\mathbf{x}_i, y_i)\}_{i=1}^{n_b}$ is the $b$-th training set with $n_b$ instances. A training

instance $\mathbf{x}_i \in \mathbb{R}^D$ belongs to class $y_i \in Y_b$. $Y_b$ is the label space of task $b$, and $Y_b \cap Y_{b'} = \varnothing$ for $b \neq b'$. Following (Rebuffi et al., 2017), a fixed number of *exemplars* from the former classes are selected as the exemplar set $\mathcal{E}$. During the $b$-th incremental stage, we can only access data from $\mathcal{D}^b$ and $\mathcal{E}$ for model training. The target is to build a unified classifier for all seen classes $\mathcal{Y}_b = Y_1 \cup \cdots Y_b$ continually. In other words, we aim to find a model $f(\mathbf{x}) : X \to \mathcal{Y}_b$ that minimizes the expected risk:

$$f^* = \underset{f \in \mathcal{H}}{\operatorname{argmin}} \, \mathbb{E}_{(\mathbf{x},y) \sim \mathcal{D}_t^1 \cup \cdots \mathcal{D}_t^b} \mathbb{I}\left(y \neq f(\mathbf{x})\right) , \tag{1}$$

where $\mathcal{H}$ denotes the hypothesis space and $\mathbb{I}(\cdot)$ is the indicator function. $\mathcal{D}_t^b$ denotes the data distribution of task $b$. In this paper, we consider CIL as a main task and also consider other continual learning scenarios like continual cross-modal retrieval in Section 5.4. Following (Wang et al., 2022c;d;b), we assume that a pre-trained vision-language model (*e.g.*, CLIP (Radford et al., 2021)) is available as the initialization for $f(\mathbf{x})$. During pre-training, CLIP jointly learns an image encoder $g_i(\cdot) : \mathbb{R}^D \to \mathbb{R}^d$ and a text encoder $g_t(\cdot) : \mathbb{R}^{Dt} \to \mathbb{R}^d$ in a contrastive manner, where $D/Dt$ are input dimensions of image/text, and $d$ is the embedding dimension. CLIP projects a batch of image-text pairs into a shared embedding space. It maximizes the cosine similarity of paired inputs and minimizes it for unmatched ones. Benefiting from the massive training data, CLIP can synthesize a *zero-shot classifier* that generalizes to unseen classes. The output of CLIP is formulated as follows:

$$p(y_i \mid \mathbf{x}) = \frac{\exp\left(\cos\left(\mathbf{z}, \mathbf{w}_i\right)/\tau\right)}{\sum_{j=1}^{|\mathcal{Y}_b|} \exp\left(\cos\left(\mathbf{z}, \mathbf{w}_j\right)/\tau\right)} , \tag{2}$$

where $\cos(\cdot, \cdot)$ denotes cosine similarity, $\tau$ is learnable temperature parameter, $\mathbf{z} = g_i(\mathbf{x})$ is the image embedding. Correspondingly, $\mathbf{w}_i$ is the text embedding of class $y_i$ obtained by feeding templated texts, *e.g.*, "a photo of a [CLASS]" into the text encoder. We denote the templated text of class $i$ as $\mathbf{t}_i$. Eq. 2 aims to find the most similar text $\mathbf{t}_i$ that maximizes the cosine similarity to the query image.

### 3.2 Overcome Forgetting in Class-Incremental Learning

**Vision-Based Learning:** CIL methods primarily rely on the *image encoder* to capture new patterns. L2P (Wang et al., 2022d) leverages visual prompt tuning (Jia et al., 2022) to enable incremental updates of a pre-trained Vision Transformer (Dosovitskiy et al., 2020). By keeping the image encoder frozen, L2P trains a learnable prompt pool and combines it with patch embeddings to obtain instance-specific embeddings. By freezing the encoder, L2P grasps the new pattern with limited forgetting. **CLIP Tuning:** The issue of tuning VLM without forgetting in CIL remains unaddressed, as previous works have solely focused on transferring CLIP to downstream tasks without considering the performance of former tasks. CoOp (Zhou et al., 2022b) converts text template into a learnable prompt, *i.e.*, $\mathbf{t}_i = [V]_1[V]_2 \cdots [V]_M[CLASS]_i$. With learned prompt, it enables the model to be transferred to the downstream task. However, since the prompt is shared for all tasks, sequentially tuning CoOp will suffer forgetting of former concepts.

**Discussions:** Current methods focus on different aspects of CIL. Vision-based methods address the issue of forgetting but neglect the valuable semantic information conveyed in texts. Conversely, CLIP's pre-trained text encoder captures class-wise relationships that can enhance model learning. Meanwhile, transfer learning methods effectively leverage the cross-modal information while sequentially tuning them suffers the catastrophic forgetting of former concepts. Is it possible to combine the cross-modal information and meanwhile resist catastrophic forgetting?

## 4 Proof: Projection Fusion for VLM

Observing the limitations of typical vision-based methods in utilizing textual information and forgetting in CLIP tuning, we aim to leverage cross-modality knowledge in CLIP while effectively mitigating forgetting. We first make the model *retentive* to learn without forgetting by learning projections to map the pre-trained features in the projected feature space. Our unique training strategy ensures the preservation of former knowledge by freezing old projections and expanding new ones for new tasks. Besides, we also make the model *comprehensive* by co-adapting and utilizing cross-modal information to enhance unified predictions. The query instance's embedding is influenced by both visual and textual information, allowing for instance-specific adaptation and enabling comprehensive predictions. In the following sections, we introduce the learning paradigm and the co-adaptation process. Lastly, we provide detailed guidelines for training and inference.

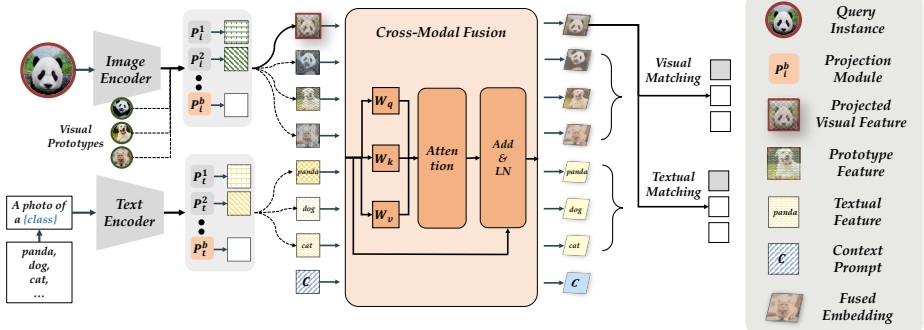

Figure 1: Illustration of PROOF. The model learns expandable projections and aggregates them to get the aggregated features. The query instance, prototype features, textual features, and context prompts are fed into the cross-modal fusion module. The fusion process utilizes self-attention to co-adapt the input set, which outputs the adapted features. The adapted query embedding is separately matched among the visual prototypes and textual features to get the final prediction. Red parts are trainable while gray ones are frozen.

## 4.1 EXPANDABLE FEATURE PROJECTION

CLIP is known for its strong zero-shot performance, *i.e.*, Eq. 2 obtains competitive results even without explicit training on the specific tasks. However, given the domain gap between pre-trained and downstream tasks, an *adaptation* process is still necessary to capture the characteristics of the latter. Specifically, we introduce a *linear layer* (denoted as "**projection**"), which is appended after the frozen image and text embeddings to facilitate the matching of pair-wise projected features. Denoting the projection of image/text as $P_i(\cdot): \mathbb{R}^d \to \mathbb{R}^d$ and $P_t(\cdot): \mathbb{R}^d \to \mathbb{R}^d$, Eq. 2 is transformed into:

$$p(y_i \mid \mathbf{x}) = \underbrace{\frac{\exp\left(\cos\left(P_i\left(\mathbf{z}\right), P_t\left(\mathbf{w}_i\right)\right)/\tau\right)}{\sum_{j=1}^{|\mathcal{Y}_b|} \exp\left(\cos\left(P_i\left(\mathbf{z}\right), P_t\left(\mathbf{w}_j\right)\right)/\tau\right)}}_{\text{Projected Matching}} . \tag{3}$$

We denote the classification based on Eq. 3 as $f_{\mathbf{PM}}(\mathbf{x})$. By freezing the image and text encoders, it aligns the downstream features in the projected space, allowing the model to encode the relevant downstream information into projection layers. Since the pre-trained model outputs generalizable features, the projection layer learns to *recombine* features in a data-driven manner. For instance, in a task involving 'birds,' the projection would assign a higher weight to features like 'beaks' and 'wings.' This adaptation enables the projected features to better discern and recognize downstream tasks.

**Expandable Projections:** However, sequentially training a *single* projection layer still leads to forgetting of former tasks, resulting in confusion when combining old and new concepts. To this end, we expand *task-specific* projections for each new task. Specifically, we append a newly initialized projection layer $P_i^b, P_t^b$ when a new task $\mathcal{D}^b$ arrives. This results in a set of projections: $\{P_i^1, P_i^2, \cdots P_i^b,\}, \{P_t^1, P_t^2, \cdots P_t^b,\}$, and we adopt the *aggregation* as the output, *i.e.*,

$$P_i(\mathbf{z}) = \sum_{m=1}^{b} P_i^m(\mathbf{z}), \quad P_t(\mathbf{w}) = \sum_{n=1}^{b} P_t^n(\mathbf{w}) . \tag{4}$$

In Eq. 4, projected features from different stages are mapped and aggregated to capture the different emphases of former and latter tasks. For example, former tasks might emphasize 'beak' features for bird recognition, while later tasks may focus on 'beard' features to differentiate cats. The aggregation of different projections produces a comprehensive representation of the query instance. By substituting Eq. 4 into Eq. 3, the model aligns the unified features in the joint space.

**How to resist forgetting of former projections?** To overcome forgetting old concepts, we freeze the projections of former tasks when learning new ones, *i.e.*, $\{\bar{P}_i^1, \bar{P}_i^2, \cdots P_i^b,\}$ (same for $P_t$). It allows the newly initialized projection to learn the *residual* information of new tasks, incorporating new concepts while preserving the knowledge of former ones. During the learning process of task $b$, we optimize the cross-entropy loss to encode the task-specific information into the current projections.

**Effect of projections**: The illustration of projections are shown in Figure 1 (left). PROOF learns projections based on the pre-trained encoders, which fits new patterns and maintains the generalizability of the pre-trained model. The parameter number of each projection layer is $d \times d$, which is negligible for the pre-trained model. These projections can be further merged during inference to alleviate the storage budget, as discussed in Section D.2. Furthermore, the model learns new projections for new tasks, and task-specific projections fit new concepts easily. Since we only optimize the current projections and freeze old ones, the former knowledge is preserved, and forgetting is alleviated.

## 4.2 Contextualizing Projections with Projection Fusion

In Eq. 3, the projected visual and textual features (*i.e.*, $\mathbf{z}$ and $\mathbf{w}$) are directly matched in the joint space. However, it would be beneficial to further *refine* these features to capture the *contextual relationship* between images and texts. For instance, when the query instance is a 'panda,' it is desirable to adjust the visual features $\mathbf{z}$ to highlight the discriminative attributes such as *black eyes and ears*. Meanwhile, considering the visual embedding of a panda, the textual features $\mathbf{w}$ should also be adapted in a *coherent* manner so that co-adapted visual and textual feature can lead to more discriminative predictions. Similarly, when the query instance is a 'cat,' features like beards and tails should be emphasized jointly for visual and textual embeddings. This adjustment process involves jointly adapting the query embedding and the context (*e.g.*, textual information) to obtain a *contextualized* embedding. A desirable adjustment function should be able to *relate* every other component as context to conduct joint adaptation. Correspondingly, we propose a *set-to-set* function that contextualizes and fuses the query embeddings and contextual information.

Specifically, we denote the adaptation function as $\mathcal{T}(\cdot)$. It receives the query instance and context information as bags, *i.e.*, $[P_i(\mathbf{z}), \mathbf{Context}]$, and outputs the set of adjusted embeddings while being permutation-invariant: $\mathcal{T}([P_i(\mathbf{z}), \mathbf{Context}]) = [\tilde{P_i(\mathbf{z})}, \tilde{\mathbf{Context}}]$. $\mathcal{T}(\cdot)$ encodes the set information and performs adaptation on each component. In the following, we describe the construction of the context information $\mathbf{Context}$ and provide details on the implementation of the set-to-set function.

**How to define the context?** Context should be items that have the potential to influence the visual embedding for more discriminative prediction. In Eq. 3, the mapping is established between the query instance and the textual information (*i.e.*, classifiers). The classifiers represent the typical textual description of the corresponding class, *i.e.*, the common feature. Hence, a naïve idea is to utilize textual features as the context, *i.e.*, $\mathbf{Context} = \mathbf{W}$, $\mathbf{W} = [P_t(\mathbf{w}_1), P_t(\mathbf{w}_2), \cdots, P_t(\mathbf{w}_{|\mathcal{Y}_b|})] \in \mathbb{R}^{|\mathcal{Y}_b| \times d}$ is the concatenation of all textual classifiers. However, (Liang et al., 2022) finds an inherent domain gap between the visual and textual embeddings in VLM. The gap leads to visual and textual embeddings residing in two separate clusters in the embedding space, which hinders effective pairwise mapping. Correspondingly, we leverage visual prototype features (Snell et al., 2017) as a useful tool for capturing the common characteristics of each class. Define the *visual prototype* of class $k$ as: $\mathbf{p}_k = \frac{1}{N} \sum_{j=1}^{|\mathcal{D}^b|} \mathbb{I}(y_j = k) g_i(\mathbf{x}_j)$, where $N = \sum_{j=1}^{|\mathcal{D}^b|} \mathbb{I}(y_j = k)$. They are calculated via forward pass at the beginning of each incremental stage and stay fixed in subsequent tasks. Visual prototypes are *representative* features of the corresponding class, which can serve as the *visual context* to adjust the embeddings. Hence, we augment the context with projected visual information, *i.e.*, $\mathbf{Context} = [\mathbf{P}, \mathbf{W}]$, where $\mathbf{P} = [P_i(\mathbf{p}_1), P_i(\mathbf{p}_2), \cdots, P_i(\mathbf{p}_{|\mathcal{Y}_b|})] \in \mathbb{R}^{|\mathcal{Y}_b| \times d}$ is the concatenation of all visual prototypes. Combining prototypes from multiple modalities helps the model adapt and fuse information in a *cross-modal* manner, which goes beyond simple visual-textual matching.

**Implementing $\mathcal{T}(\cdot)$ with Self-Attention:** Given the above context information, the design of cross-modal fusion should be able to influence instance embedding with the context (*e.g.*, highlighting the visual features like black eyes and ears for a panda), and vice versa. To this end, we use the self-attention (SA) mechanism (Vaswani et al., 2017; Lin et al., 2017) as the cross-modal fusion function $\mathcal{T}(\cdot)$. Being permutation invariant, self-attention is good at outputting adapted embeddings even with long dependencies, which naturally suits the characteristics of the adaptation function. Specifically, SA takes the triplets (query $\mathcal{Q}$, key, $\mathcal{K}$, and value $\mathcal{V}$) as input. The inputs are projected into the same embedding space, *i.e.*, $K = W_K^\top [\mathbf{k}_k; \forall \mathbf{k}_k \in \mathcal{K}] \in \mathbb{R}^{d \times |\mathcal{K}|}$. Similar projections are made for the query $\mathcal{Q}$ and value $\mathcal{V}$. Afterward, the query $\mathbf{x}_q \in \mathcal{Q}$ is matched against a list of keys $K$ where each key has a value $V$. The output of self-attention is the sum of all the values weighted by the proximity of the key to the query point:

$$\tilde{P}_i(\mathbf{z}) = P_i(\mathbf{z}) + \sum_k \alpha_{qk} V_{:,k} \,, \tag{5}$$

where $\alpha_{qk} \propto \exp\left(\frac{P_i(\mathbf{z})^\top W_Q \cdot K}{\sqrt{d}}\right)$, $V_{:,k}$ is the $k$-th column of $V$. In Eq. 5, the fused visual feature $\tilde{P}_i(\mathbf{z})$ adds the attention part based on the original feature $P_i(\mathbf{z})$. Since $\alpha_{qk}$ reflects the relative similarity of the visual feature to other components, the second term in Eq. 5 adjusts the input by considering its relationship to other components. Apart from the visual feature, the adaptation process is the same for other components in $\mathbf{Context}$. Specifically, we have $\mathcal{Q} = \mathcal{K} = \mathcal{V} = [P_i(\mathbf{z}), \mathbf{Context}]$.

**Effect of Cross-Modal Fusion**: We illustrate cross-modal fusion in Figure 1 (right). Since we utilize the visual and textual information of seen classes as context information, we can make the visual embedding *instance-specific* and discriminative. The fusion model is trained incrementally to adjust embeddings to reflect the context information as data evolves. With the contextualized embeddings, we can conduct the *visual mapping* and *textual matching*:

$$p(y_i \mid \mathbf{x}) = \underbrace{\frac{\exp\left(\cos\left(\tilde{P}_i(\mathbf{z}), \tilde{P}_i(\mathbf{p}_i)\right)/\tau\right)}{\sum_{j=1}^{|\mathcal{Y}_b|} \exp\left(\cos\left(\tilde{P}_i(\mathbf{z}), \tilde{P}_i(\mathbf{p}_j)\right)/\tau\right)}}_{\text{Visual Matching}} + \underbrace{\frac{\exp\left(\cos\left(\tilde{P}_i(\mathbf{z}), \tilde{P}_t(\mathbf{w}_i)\right)/\tau\right)}{\sum_{j=1}^{|\mathcal{Y}_b|} \exp\left(\cos\left(\tilde{P}_i(\mathbf{z}), \tilde{P}_t(\mathbf{w}_j)\right)/\tau\right)}}_{\text{Textual Matching}} . \quad (6)$$

In Eq. 6, we consider two matching targets for the fused visual feature, *i.e.*, visual and textual matching. Visual matching assigns logits by the similarity to the adapted visual prototypes, while textual matching does it for textual features. Hence, we are able to achieve better prediction performance by such cross-modal matching process.

**Learning Context Prompts**: In addition to visual prototypes and textual classifiers, we also introduce a set of learnable *context prompts* $\{\mathbf{c}^1, \cdots, \mathbf{c}^b\}, \mathbf{c}^i \in \mathbb{R}^{c \times d}$ to be optimized as data evolves. $c$ denotes the length of each prompt. Similar to projection layers, we make the context prompts *expandable* to catch the new characteristics of new tasks. We initialize a new context prompt while learning a new task and freeze others $\{\bar{\mathbf{c}}^1, \bar{\mathbf{c}}^2, \cdots, \mathbf{c}^b\}$. Freezing former context prompts enables us to catch new task-specific features and meanwhile preserving former ones. The context prompts serve as *adaptable* context information, enhancing the co-adaption. Afterward, the context information is formulated as **Context** $= [\mathbf{P}, \mathbf{W}, \mathbf{C}]$, where $\mathbf{C}$ is the aggregation of all context prompts. As shown in Figure 1, context prompts $\mathbf{C}$ only encode the task-specific information into the self-attention process, and they not serve as the matching target in Eq. 6.

### 4.3 SUMMARY OF PROOF

In PROOF, we first enable learning new concepts via projected mapping. Then, to accommodate new concepts without interference from previous ones, we initialize new projections for each new task and freeze the former ones. Besides, we utilize self-attention to adjust the embeddings of the query instance and the context information to promote cross-modal fusion. Figure 1 illustrates three predictions, *i.e.*, projected matching (Eq. 3), visual/textual matching (Eq. 6). We denote these models as $f_{\text{PM}}(\mathbf{x}), f_{\text{VM}}(\mathbf{x}), f_{\text{TM}}(\mathbf{x})$, respectively. During training, we optimize the cross-entropy loss:

$$\min_{\{P_i^b, P_t^b, \mathcal{T}, \mathbf{c}^b\}} \ell(f_{\text{PM}}(\mathbf{x}), y) + \ell(f_{\text{VM}}(\mathbf{x}), y) + \ell(f_{\text{TM}}(\mathbf{x}), y), \quad (7)$$

where $(\mathbf{x}, y) \in \mathcal{D}^b \cup \mathcal{E}$. In Eq. 7, all pre-trained weights are frozen, and we only optimize these *additional* parameters. For inference, we aggregate the three logits, *i.e.*, $f(\mathbf{x}) = f_{\text{PM}}(\mathbf{x}) + f_{\text{VM}}(\mathbf{x}) + f_{\text{TM}}(\mathbf{x})$. We give the pseudo-code of PROOF to illustrate the training/inference process in Section A.

## 5 EXPERIMENT

In this section, we compare PROOF in comparison to state-of-the-art methods on benchmark datasets to investigate the capability of overcoming forgetting. Besides, we conduct ablations to analyze the effect of each component in the model. We also extend PROOF to other VLMs and continual learning scenarios, experiment with a non-overlapping dataset, and address the zero-shot performance degradation phenomena. Further details and experimental results can be found in the supplementary.

### 5.1 EXPERIMENTAL SETUP

**Dataset**: Following the benchmark CIL settings (Rebuffi et al., 2017; Wang et al., 2022d;c; Yu et al., 2020; Zhou et al., 2023c), we evaluate the performance on **CIFAR100** (Krizhevsky et al., 2009), **CUB200** (Wah et al., 2011), **ObjectNet** (Barbu et al., 2019), and **ImageNet-R** (Hendrycks et al., 2021). We also follow the benchmark in VLM tuning (Zhou et al., 2022b), and formulate **FGVCAircraft** (Maji et al., 2013), **StanfordCars** (Krause et al., 2013), **Food101** (Bossard et al., 2014), **SUN397** (Xiao et al., 2010) and **UCF101** (Soomro et al., 2012) into CIL setting. Specifically, we sample (a subset of) 100 classes from CIFAR100, Aircraft, Cars, Food, UCF, 200 classes from

Table 1: Average and last performance of different methods. The first and second columns represent the methods with and without exemplars. Full results are reported in Section D.11. **All methods are initialized with the same pre-trained CLIP for a fair comparison (see Section E.3).**

| Method | Exemplar | ImageNet-R | | | | CUB | | | | UCF | | | |
|---|---|---|---|---|---|---|---|---|---|---|---|---|---|
| | | B0 Inc20 | | B100 Inc20 | | B0 Inc20 | | B100 Inc20 | | B0 Inc10 | | B50 Inc10 | |
| | | $\bar{\mathcal{A}}$ | $\mathcal{A}_B$ | $\bar{\mathcal{A}}$ | $\mathcal{A}_B$ | $\bar{\mathcal{A}}$ | $\mathcal{A}_B$ | $\bar{\mathcal{A}}$ | $\mathcal{A}_B$ | $\bar{\mathcal{A}}$ | $\mathcal{A}_B$ | $\bar{\mathcal{A}}$ | $\mathcal{A}_B$ |
| Finetune | ✗ | 1.37 | 0.43 | 1.01 | 0.88 | 2.06 | 0.64 | 0.56 | 0.47 | 4.51 | 1.59 | 1.21 | 0.80 |
| Finetune LiT (Zhai et al., 2022) | ✗ | 64.88 | 30.42 | 57.75 | 29.77 | 58.15 | 35.28 | 51.95 | 35.96 | 79.25 | 64.84 | 81.79 | 65.4 |
| Finetune CoOp (Zhou et al., 2022b) | ✗ | 60.73 | 37.52 | 54.20 | 39.77 | 27.61 | 8.57 | 24.03 | 10.14 | 47.85 | 33.46 | 42.02 | 24.74 |
| SimpleCIL (Zhou et al., 2023c) | ✗ | 81.06 | 74.48 | 76.84 | 74.48 | 83.81 | 77.52 | 79.75 | 77.52 | 90.44 | 85.68 | 88.12 | 85.68 |
| ZS-CLIP (Radford et al., 2021) | ✗ | 83.37 | 77.17 | 79.57 | 77.17 | 74.38 | 63.06 | 67.96 | 63.06 | 75.50 | 67.64 | 71.44 | 67.64 |
| CoOp (Zhou et al., 2022b) | ✓ | 82.40 | 76.20 | 79.76 | 77.13 | 77.34 | 68.70 | 74.09 | 67.47 | 90.13 | 86.24 | 88.36 | 85.71 |
| iCaRL (Rebuffi et al., 2017) | ✓ | 72.22 | 54.38 | 68.67 | 60.15 | 82.04 | 74.74 | 78.57 | 75.07 | 89.47 | 84.34 | 88.51 | 84.11 |
| LUCIR (Hou et al., 2019) | ✓ | 72.44 | 55.12 | 68.66 | 60.12 | 82.64 | 75.93 | 76.47 | 74.93 | 89.38 | 84.68 | 88.06 | 85.61 |
| DER (Yan et al., 2021) | ✓ | 80.21 | 73.47 | 75.91 | 73.05 | 77.10 | 65.48 | 71.68 | 65.56 | 84.86 | 75.12 | 83.99 | 77.05 |
| MEMO (Zhou et al., 2023b) | ✓ | 80.00 | 74.07 | 76.72 | 73.95 | 77.32 | 65.69 | 72.88 | 66.41 | 84.02 | 74.08 | 82.58 | 75.48 |
| L2P (Wang et al., 2022d) | ✓ | 75.73 | 67.22 | 74.15 | 71.20 | 79.23 | 68.54 | 75.85 | 71.12 | 88.71 | 83.93 | 86.51 | 83.22 |
| DualPrompt (Wang et al., 2022c) | ✓ | 78.47 | 70.82 | 72.98 | 69.18 | 83.21 | 74.94 | 78.06 | 74.27 | 89.48 | 85.41 | 86.96 | 84.65 |
| PROOF | ✓ | **85.34** | **80.10** | **82.32** | **80.30** | **84.93** | **79.43** | **81.67** | **79.18** | **92.34** | **89.92** | **91.70** | **89.16** |

Figure 2: Incremental performance of different methods. We report the performance gap after the last incremental stage of PROOF and the runner-up method at the end of the line. Finetune-based methods in Table 1 are not plotted due to their inferior performance. **All methods are based on the same backbone and weight.**

CUB200, ObjectNet, ImageNet-R, and 300 classes from SUN to ease the data split. Following (Rebuffi et al., 2017), the training class order is shuffled with random seed 1993. The dataset splits are denoted as **Base-$x$, Inc-$y$**, where $x$ represents the number of classes in the first stage, and $y$ represents the number of new classes in each subsequent task. $x = 0$ means each task contains $y$ classes.

**Comparison methods:** We first compare to SOTA CIL methods iCaRL (Rebuffi et al., 2017), LUCIR (Hou et al., 2019), MEMO (Zhou et al., 2023b), DER (Yan et al., 2021) SimpleCIL (Zhou et al., 2023c) L2P (Wang et al., 2022d), DualPrompt (Wang et al., 2022c). Denote the baseline of sequential finetuning as Finetune; we combine it with different tuning techniques, *e.g.*, LiT (Zhai et al., 2022) and CoOp (Zhou et al., 2022b). We also report the zero-shot performance of CLIP as ZS-CLIP by matching the query instance to the template (Eq. 2). All methods are based on the **same pre-trained CLIP for fair comparison.**

**Implementation details:** We deploy all methods with PyTorch (Paszke et al., 2019) on Tesla V100. We use the *same* network backbone, *i.e.*, CLIP with ViT-B/16 for all compared methods for *fair comparison*. We experiment with two commonly used pre-trained CLIP weights, *i.e.*, OpenAI (Radford et al., 2021) and OpenCLIP LAION-400M (Ilharco et al., 2021). The model is trained with a batch size of 64 for 5 epochs, and we use SGD with momentum for optimization. The learning rate starts from 0.001 and decays with cosine annealing. Following (Rebuffi et al., 2017), we use the herding (Welling, 2009) algorithm to select 20 exemplars per class for rehearsal. We report the performance of different exemplar scale in Section E.5. The context prompt length is set to 3, and the head of self-attention is set to 1. The template for classification is the same as (Mu et al., 2022). The source code will be made publicly available upon acceptance.

**Evaluation Metrics:** Denote the accuracy after the $b$-th stage as $\mathcal{A}_b$, we follow (Rebuffi et al., 2017)

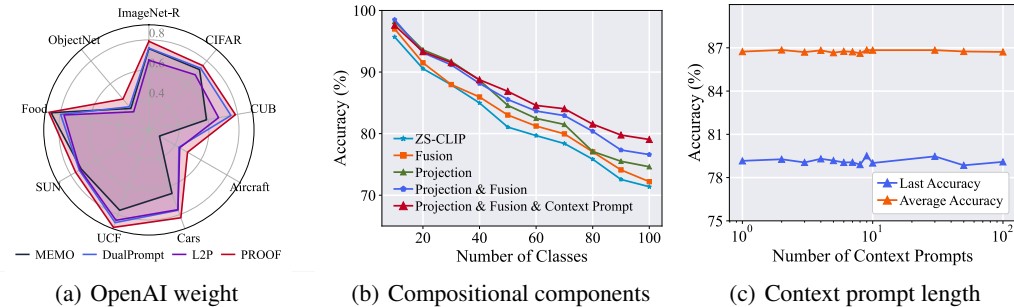

| (a) OpenAI weight | (b) Compositional components | (c) Context prompt length |

Figure 3: Ablation study. **Left:** experiments on nine benchmarks with OpenAI weights. **Middle:** ablation study on compositional components in PROOF. Every part improves the performance of CIL. **Right:** $\mathcal{A}_B$ and $\bar{\mathcal{A}}$ with change of context prompts. The performance is robust to the change of context prompt length.

to use $\mathcal{A}_B$ (last stage performance) and $\bar{\mathcal{A}} = \frac{1}{B} \sum_{b=1}^{B} \mathcal{A}_b$ (average performance) for evaluation. We report the forgetting measure (Wang et al., 2023a) of different methods in Section E.4.

## 5.2 BENCHMARK COMPARISON

We report the results on nine benchmark datasets using CLIP with ViT-B/16 (OpenCLIP LAION-400M) in Table 1 and Figure 2. These splits include the scenarios with large and small base classes. Notably, PROOF consistently achieves the best performance among all the methods compared. Sequential finetuning of the model with contrastive loss leads to significant forgetting, irrespective of the tuning techniques employed (*e.g.*, LiT and CoOp). Since SimpleCIL and ZS-CLIP do not finetune the model parameters, they achieve competitive results by transferring the knowledge from the pre-training stage into the downstream tasks. However, most methods achieve better performance than ZS-CLIP, indicating the importance of incremental learning on downstream tasks.

Specifically, we can draw three key conclusions from these results. 1) The first stage performance of PROOF surpasses that of the typical prompt learning method, CoOp, thus validating the effectiveness of learning projections for downstream tasks. 2) The performance curve of PROOF consistently ranks at the top across all methods, demonstrating its capability to resist forgetting. 3) Compared to vision-only methods (*i.e.*, L2P and DualPrompt), PROOF exhibits substantial improvement, indicating textual and visual information can be co-adapted to facilitate incremental learning. We report the running time comparison of different methods in Section D.3.

## 5.3 ABLATION STUDY

**Different backbone weights:** The comparison in Section 5.2 is based on LAION-400M pre-trained CLIP. As another popular pre-trained weight, we also explore the performance of the weights provided by OpenAI. We report the last accuracy $\mathcal{A}_B$ of four competitive methods on nine benchmarks in Figure 3(a). We report the full results of the incremental performance in Section D.6. As depicted in the figure, PROOF still performs the best on all datasets among all compared methods.

**Compositional components:** We experiment on CIFAR100 B0 Inc10 to investigate the importance of each part in PROOF. Specifically, we compare the performance of PROOF and its sub-modules, *i.e.*, projections and cross-modal fusion. The results, shown in Figure 3(b), indicate that training expandable projections or the fusion module individually can both enhance the performance of vanilla CLIP. This suggests that the expandable task representation and cross-modal information can help the learning process. Furthermore, when combining them together, we find 'Projection & Fusion' further show better performance than any of them, verifying that they can work together by fusing the expandable representations. Lastly, when incorporating the context prompts, the model shows the best performance among all variations, verifying the effectiveness of expandable task-specific prompts in incremental learning. Ablations verify the importance of each component in PROOF.

**Number of context prompts:** Figure 3(b) verifies the strong performance of context prompts, and we explore the appropriate length $c$ of the context prompt on CIFAR100 B0 Inc10. By varying the number of $c$ among $\{1, 2, 3, 4, 5, 6, 7, 8, 9, 10, 30, 50, 100\}$, we report the average performance and last performance of PROOF in Figure 3(c). As shown in the figure, the performance of PROOF is robust with the change of the prompt length, and we set $c = 3$ as the default length.

## 5.4 FURTHER ANALYSIS

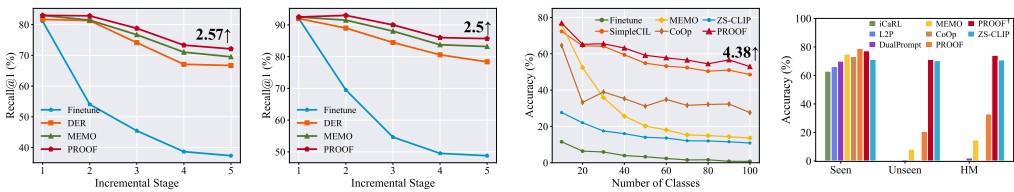

(a) Flickr30K IR@1    (b) Flickr30K TR@1    (c) TV100 Base0 Inc10    (d) $\mathcal{A}_S$, $\mathcal{A}_U$, $\mathcal{A}_{HM}$

Figure 4: **(a)(b):** experiments on continual cross-modal retrieval task with BEiT-3. **(c):** experiments on TV100, a non-overlapping dataset to pre-trained CLIP. **(d):** accuracy of seen, unseen, and harmonic mean (HM) at the last incremental stage. PROOF$^{\dagger}$ strikes a balance between adaptivity and the ZS performance.

**Extending PROOF to other VLMs and other continual learning scenarios:** As defined in Eq. 4, PROOF adds expandable projections on the VLM to trace task-specific features for continual learning. Hence, PROOF is a general algorithm that does not rely on the CLIP structure or the CIL setting. Correspondingly, we verify PROOF in the **continual cross-modal retrieval (CCMR) setting** with BEiT-3 (Wang et al., 2023b). The cross-modal retrieval task requires the VLM to search for the most related image given the text description, and vice versa. To construct a CCMR task, we split the Flickr30K dataset (Plummer et al., 2015) into several subsets by matching the description with several keywords, *i.e.*, {walk, stand, run, ride, play} (as shown in Figure 5). This enables us to split the large dataset into several subsets, and continually learning the new CCMR task will result in forgetting former tasks with different topics. We continually train the model with the CCMR sequence and evaluate the performance for image→text and text→image with the benchmark metric (Song & Soleymani, 2019) $R@1/5/10$. Figures 4(a),4(b) show the image and text retrieval results. We find other continual learning algorithms (DER and MEMO) face catastrophic forgetting in CCMR, while PROOF still performs competitively even with different VLMs and continual learning scenarios. We report more details about the task construction and experimental results in Section B.

**Class-incremental learning with non-overlapping dataset:** We have verified PROOF's performance on benchmark CIL datasets in Section 5.2. However, one may argue that these benchmark datasets may have data overlapping with CLIP's pre-training dataset. Hence, we manually collect a new dataset for TV series classification with TV series after 2021 (the publication of CLIP), namely TV100. TV100 contains 100 classes of posters, actor images, and stage photos of the corresponding TV series, which shares no data overlapping with CLIP's pre-training data and can be viewed as totally new knowledge to CLIP. We evaluate the CIL performance with TV100 in Figure 4(c), where PROOF still outperforms other competitors by a substantial margin. More details about dataset collection and selection are reported in Section D.10.

**Exploring Zero-Shot Performance:** Apart from the ability to learn new concepts, CLIP is also known to have a strong zero-shot (ZS) ability. However, *continuously* updating the model weakens the generalizability and harms the ZS performance on subsequent tasks. Hence, apart from evaluating 'seen' classes ($\mathcal{Y}_b = Y_1 \cup \cdots Y_b$), we also assess the performance on 'unseen' classes $\mathcal{Y}_u = Y_{b+1} \cup \cdots Y_B$ to investigate the ZS performance, *i.e.*, $\mathcal{A}_S$ (seen classes), $\mathcal{A}_U$ (unseen classes), and $\mathcal{A}_{HM}$ (harmonic mean of $\mathcal{A}_S$ and $\mathcal{A}_U$) after each task. We compare the aforementioned measures on CIFAR100 B0 Inc10. Apart from the compared methods in Section 5.2, we also report a variation of PROOF, namely PROOF$^{\dagger}$. The only difference lies in the design of the projection, where PROOF$^{\dagger}$ uses a *residual* format $P_i(\mathbf{z}) = \sum_{m=1}^{b} \left( P_i^m(\mathbf{z}) + \mathbf{z} \right)$ as the output (same for $P_t$). As shown in Figure 7(c), most compared methods lose the ZS performance as data evolves, showing poor $\mathcal{A}_U$ than ZS-CLIP. Compared to PROOF, PROOF$^{\dagger}$ sacrifices the adaptivity to maintain ZS performance, striking a balance between seen and unseen classes. Therefore, when ZS performance is essential, using PROOF$^{\dagger}$ is the preferred choice. See more details about the zero-shot performance in Section C.

## 6 CONCLUSION

Real-world learning systems necessitate the ability to continually acquire new knowledge. In this paper, we aim to equip the popular VLM with the CIL ability. Specifically, we learn the expandable projections so that visual and textual information can be aligned incrementally. This expansion technique allows for integrating new concepts without compromising previous ones. Additionally, we enforce cross-modality fusion with the self-attention mechanism, where visual and textual information are jointly adapted to produce instance-specific embeddings. Extensive experiments validate the effectiveness of our proposed PROOF in various VLMs and various continual learning scenarios. We also demonstrate that a simple variation can preserve the model's zero-shot capability.

**Limitations:** Possible limitations include the usage of exemplars, where storage constraints and privacy issues may happen. Future works include extending the model to exemplar-free scenarios.

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

# Supplementary Material

In the main paper, we present a method to prevent forgetting in vision-language models through projection expansion and fusion. The supplementary material provides additional details on the experimental results mentioned in the main paper, along with extra empirical evaluations and discussions. The organization of the supplementary material is as follows:

- Section A presents the pseudo code of PROOF, explaining the training and testing pipeline.

- Section B extends PROOF to other vision-language models, specifically BEiT-3 (Wang et al., 2023b), in the context of continual cross-modal retrieval tasks. This section demonstrates the generalizability of PROOF by successfully adapting it to various VLMs without encountering forgetting. Experiments conducted on different VLMs and tasks highlight PROOF as a *unified and versatile* framework.

- Section C explores the zero-shot performance of different class-incremental learning algorithms via different measures, *i.e.*, seen and unseen accuracy and LAION score. We also maintain PROOF's zero-shot performance with a simple modification.

- Section D reports comprehensive experimental results from the main paper, including the full results of nine benchmark datasets with two data splits, as well as the results obtained using OpenAI weights and larger backbones. Furthermore, this section includes additional ablations such as variations of projection types, results from multiple runs, and an analysis of the number of parameters. We also collect a new dataset that strictly does not overlap with the existing pre-training dataset, and provide experimental results on it.

- Sections E and F provide detailed information on the experiments, including dataset and exemplar selection details, an introduction to the compared methods, and a discussion of the broader impacts.

## A    PSEUDO CODE

In this section, we provide a detailed explanation of PROOF by presenting the pseudo-code in Alg 1. In each incremental stage, we are provided with the training dataset $\mathcal{D}^b$ and the exemplar set $\mathcal{E}$, with the objective of updating the current model $f(\cdot)$. Prior to training, we initially extract visual prototypes for the new classes (Line 1). These prototypes are calculated using the frozen visual embedding $g_i(\cdot)$, ensuring their stability throughout model updates. Subsequently, we freeze the former projections and context prompts while initializing new projections and context prompts specifically for the new incremental task (Line 2 to Line 4). These steps represent the model expansion process, which is followed by the subsequent learning process.

During the learning process, we concatenate the training instances from the current dataset and the exemplar set, initiating a for-loop. For each instance-label pair, we calculate the projected visual and textual embeddings (Line 6 to Line 9). Subsequently, we compute the projected matching loss (Line 10) to encode task-specific information into the current projection layers. Based on the projected features, we derive context information and perform cross-modal fusion (Line 11 to Line 13). Consequently, we obtain three logits for model updating and utilize the cross-entropy loss to update these modules (Line 14). The updated model is then returned as the output of the training process.

**Discussions:** Besides the simple addition operation, there exist alternative methods for aggregating information from multiple projections. However, due to the requirement of fixed input dimensionality for cross-modal fusion, we refrain from using concatenation as the aggregation function. Furthermore, it is worth noting that MEMO (Zhou et al., 2023b) can be viewed as a specific case where concatenation is employed for aggregation. Nonetheless, its inferior performance (as shown in Table 4) suggests that summation is a more favorable choice.

## B    EXTENSION TO OTHER VISION LANGUAGE MODELS

In the main paper, we use CLIP (Radford et al., 2021) as an exemplar vision-language model due to its popularity and representativeness. However, the field of vision-language models is rapidly advancing,

---

**Algorithm 1** Training PROOF for CIL

---

**Input**: Training dataset: $\mathcal{D}^b$; Exemplar set: $\mathcal{E}$; Current model: $f(\cdot)$;
**Output**: Updated model;
  1: Extract prototypes $\mathbf{p}$ for each new class in $\mathcal{D}^b$;
  2: Freeze current projections and context prompts;
  3: Initialize new projections for the visual and textual branches, $P_i^b, P_t^b$;    ▷ Expand projections
  4: Initialize new context prompt $\mathbf{c}^b$;
  5: **for** $(\mathbf{x}, y) \in \mathcal{D}^b \cup \mathcal{E}$ **do**        ▷ Incremental learning
  6:     Calculate the visual embedding $\mathbf{z} = g_i(\mathbf{x})$;
  7:     Calculate the projected visual feature $P_i(\mathbf{z})$;
  8:     Calculate the textual embedding $\mathbf{w}$ of all seen classes;
  9:     Calculate the projected textual embeddings of all seen classes $P_t(\mathbf{w})$;
10:     Calculate the logits for projected matching $f_{\mathbf{PM}}(\mathbf{x})$ via Eq. 5;   ▷ Projected matching
11:     Calculate the projected visual features for all visual prototypes $\mathbf{p}$;
12:     Conduct cross-modal fusion via Eq. 7;       ▷ Cross-modal fusion
13:     Calculate the logits for visual and textual matching via Eq. 8;  ▷ Visual & textual matching
14:     Calculate the loss via Eq. 9; update the model;
      **return** the updated model;

---

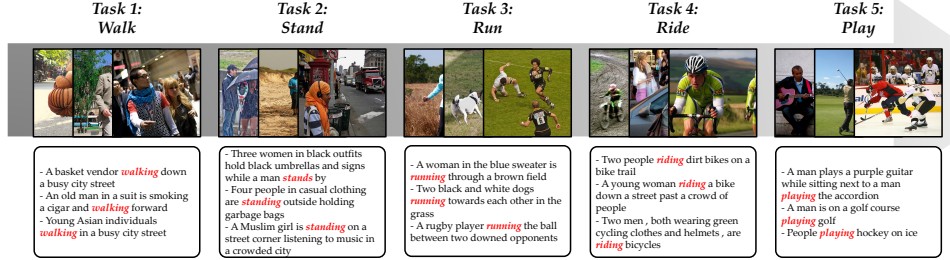

Figure 5: The training protocol of five incremental stages in Flickr30K. We split training instances into five tasks, *i.e.*, walk, stand, run, ride, and play. The training/testing sets do not include images that do not fall into these tasks. We use the pre-trained BEiT-3 as the initialization and sequentially learn cross-modal retrieval. At the end of each task, the model is evaluated on all previously learned concepts.

and various models are available. Therefore, in this section, we extend our PROOF framework to another widely used vision-language model, namely BEiT-3 (Wang et al., 2023b), focusing on the cross-modal retrieval task. BEiT-3 is a popular VLM that demonstrates promising performance across multiple vision-language tasks. When finetuning BEiT-3 for cross-modal retrieval, it functions as a *dual encoder*, similar to CLIP, featuring a dual-branch structure. As the retrieval task differs from classification, we adopt a degradation of PROOF by solely employing the projection expansion strategy without implementing cross-modal fusion. We refer the readers to the BEiT-3 paper (Wang et al., 2023b) for more details about the backbone model.

For evaluation, we employ the Flickr30K dataset (Plummer et al., 2015) to assess the performance of incremental cross-modal retrieval. Flickr30K comprises 31,783 images collected from the Flickr image-sharing platform, encompassing diverse themes such as daily life, travel, people, food, and scenes. Each image in the dataset is accompanied by *five* manually annotated textual descriptions, which provide descriptive information capturing the main content and context of the images. To formulate an incremental data stream, we utilize keyword matching to identify images containing different actions (*e.g.*, walk, stand, run, ride, play). Then, we split the training instances into five subsets based on these specific actions. Figure 5 illustrates the formulation of the stream, while images not associated with these actions are excluded from training. To create a balanced testing set,

Table 2: Average and last performance of different methods. The best is in bold. The first row stands for the text retrieval task, and the second is the image retrieval task. **All methods are based on the same backbone/weight.**

| Method | Image → Text | | | | | |
| --- | --- | --- | --- | --- | --- | --- |
| | $R_B@1$ | $\bar{R}@1$ | $R_B@5$ | $\bar{R}@5$ | $R_B@10$ | $\bar{R}@10$ |
| Finetune | 48.79 | 62.89 | 76.38 | 85.04 | 85.68 | 91.84 |
| DER (Yan et al., 2021) | 78.37 | 84.48 | 96.34 | 98.23 | 99.06 | 99.59 |
| MEMO (Zhou et al., 2023b) | 83.18 | 87.79 | 96.57 | 98.27 | 99.16 | 99.66 |
| PROOF | **85.68** | **89.43** | **97.07** | **98.68** | **99.79** | **99.86** |
| Method | Text → Image | | | | | |
| | $R_B@1$ | $\bar{R}@1$ | $R_B@5$ | $\bar{R}@5$ | $R_B@10$ | $\bar{R}@10$ |
| Finetune | 37.35 | 51.33 | 67.38 | 77.77 | 77.95 | 85.55 |
| DER (Yan et al., 2021) | 66.71 | 74.18 | 89.63 | 93.00 | 94.84 | 96.69 |
| MEMO (Zhou et al., 2023b) | 69.53 | 76.35 | 91.89 | 94.44 | 96.09 | 97.32 |
| PROOF | **72.10** | **78.01** | **93.10** | **95.27** | **96.92** | **97.90** |

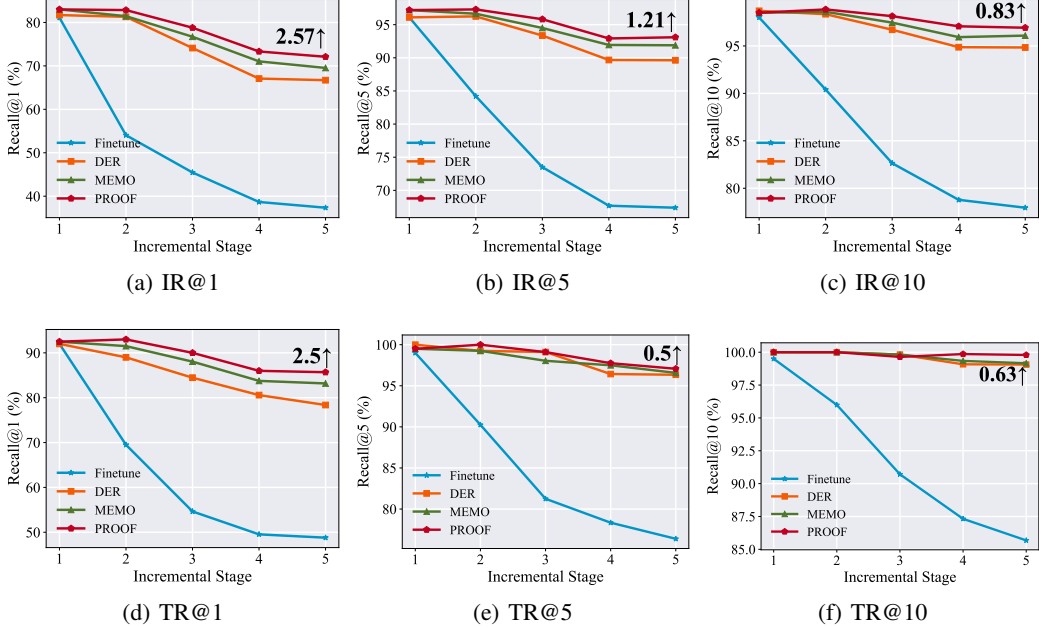

Figure 6: Incremental performance of each method. IR means the recall of image retrieval, and TR denotes the recall of text retrieval. PROOF **consistently outperforms other compared methods with a substantial margin on the incremental cross-modal retrieval task.**

we maintain a 5:1 training-to-testing ratio for splitting the training and testing pairs. Following the instructions provided by BEiT, we use 'beit3_base_itc_patch16_224[1]' as the VLM's initialization.

For evaluation, we employ standard cross-modal retrieval measures, namely $R@1$, $R@5$, and $R@10$. The retrieval is conducted in two directions: image → text and text → image. Similarly to the CIL evaluation, we also report the last recall $R_B@1$ and the average recall $\bar{R}@1$ across incremental stages. To provide a comparative analysis, we compare PROOF against typical finetuning as the baseline and modify MEMO (Zhou et al., 2023b) and DER (Yan et al., 2021) for comparison. These methods

---

[1] https://github.com/microsoft/unilm/blob/master/beit3/README.md#flickr30k-retrieval

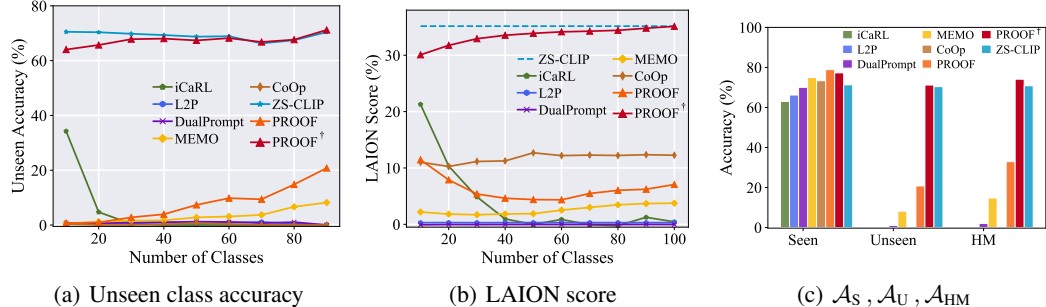

(a) Unseen class accuracy     (b) LAION score     (c) $\mathcal{A}_S$, $\mathcal{A}_U$, $\mathcal{A}_{HM}$

Figure 7: Experiment on zero-shot performance. **Left:** accuracy on unseen classes during incremental learning. **Middle:** LAION score during incremental learning. **Right:** accuracy of seen, unseen, and harmonic mean (HM) at the last incremental stage. PROOF$^\dagger$ strikes a balance between adaptivity and the ZS performance.

represent state-of-the-art CIL approaches that can be adapted with minor modifications to the current task. However, methods such as L2P and DualPrompt are unsuitable for cross-modal retrieval tasks as they do not focus on cross-modal matching.

The experimental results are presented in Table 2, and the incremental performance of each measure is depicted in Figure 6. As evident from these figures, finetuning the model with new concepts leads to catastrophic forgetting in cross-modal retrieval tasks. However, equipping the model with incremental learning abilities alleviates forgetting. Among all the compared methods, PROOF consistently achieves the best performance across different retrieval tasks and metrics, thereby verifying its effectiveness in mitigating forgetting in VLMs. Experiments conducted on different VLMs and tasks establish PROOF as a *unified and general* framework. Future work involves extending PROOF to other VLMs and applications, such as image captioning (Vinyals et al., 2015) and VQA (Antol et al., 2015).

## C  EXPLORING ZERO-SHOT PERFORMANCE

CLIP is known to have the zero-shot (ZS) ability, *i.e.*, even if the model has not been trained for recognizing the image, it can still predict the possibility of an image $\mathbf{x}$ belonging to the class $y$ by matching the cosine similarity via Eq. 2. The strong generalizability of CLIP makes it a popular model in computer vision. However, in CIL, the model is *continuously* updated with the downstream task, which weakens the generalizability and harms the ZS performance (Wortsman et al., 2022) on subsequent tasks. In this section, we explore the ZS performance degradation of CLIP and propose a variation of PROOF to maintain the ZS performance.

**Evaluation protocol for ZS performance:** Current CIL methods focus on evaluating 'seen' classes, *i.e.*, evaluating $\mathcal{Y}_b = Y_1 \cup \cdots Y_b$ after learning task $b$. However, since CLIP exhibits ZS performance, we can also assess the performance on 'unseen' classes $\mathcal{Y}_u = Y_{b+1} \cup \cdots Y_B$ to investigate the ZS performance. Correspondingly, we can obtain the performance metrics $\mathcal{A}_S$ (seen classes), $\mathcal{A}_U$ (unseen classes), and $\mathcal{A}_{HM}$ (harmonic mean of $\mathcal{A}_S$ and $\mathcal{A}_U$) after each task. Additionally, based on the LAION400M (Schuhmann et al., 2021) pre-trained CLIP, we also utilize a subset of 10,000 image-text pairs from LAION400M, and calculate the matching score of them, *i.e.*, cosine similarity of image-text embeddings. We denote the average matching score as *LAION score*, which indicates the matching degree of the adapted model on the *upstream* tasks. Given the relationship between generalizability and the upstream task, the LAION score serves as an effective measure of ZS performance.

**Results:** We compare the aforementioned measures on CIFAR100 B0 Inc10. Apart from the compared methods in Section 5.2, we also report a variation of PROOF, namely PROOF$^\dagger$. The only difference lies in the design of the projection, where PROOF$^\dagger$ uses a *residual* format $P_i(\mathbf{z}) = \sum_{m=1}^{b} (P_i^m(\mathbf{z}) + \mathbf{z})$ as the output (same for $P_t$). To investigate the ZS performance as model updates, we show the accuracy on unseen classes $\mathcal{A}_U$ along incremental stages in Figure 7(a), where ZS-CLIP shows the best performance. Due to the incorporation of pre-trained information into the projected features, PROOF$^\dagger$ maintains competitive ZS performance. Conversely, other methods experience a decline in ZS performance as their focus shifts to downstream tasks. We observe a similar trend in Figure 7(b), where PROOF$^\dagger$ achieves a LAION score similar to that of ZS-CLIP. Lastly, we report $\mathcal{A}_S$, $\mathcal{A}_U$, $\mathcal{A}_{HM}$ in the last incremental stage in Figure 7(c). We can infer a *trade-off* between the adaptivity on downstream tasks and the generalizability of ZS performance. Compared to PROOF, PROOF$^\dagger$ sacrifices the adaptivity to maintain ZS performance, striking a balance between

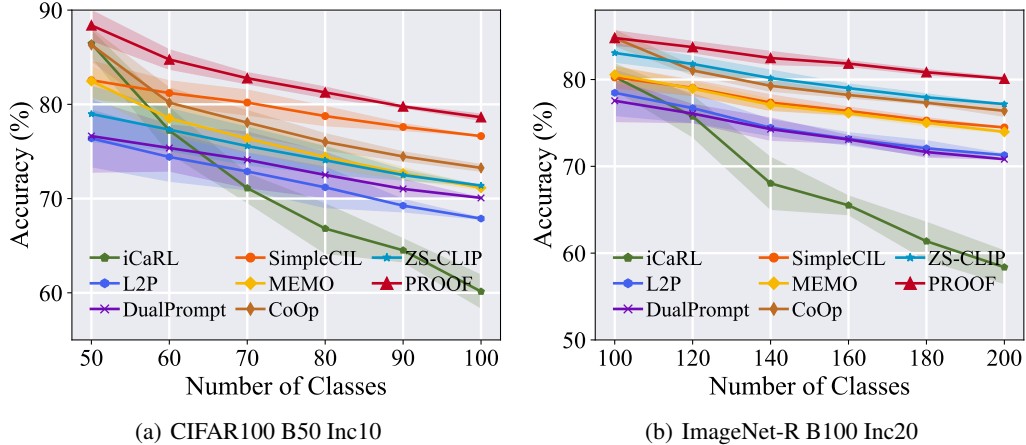

(a) CIFAR100 B50 Inc10    (b) ImageNet-R B100 Inc20

Figure 8: Results of multiple runs for CIFAR100 and ImageNet-R. The solid line represents the mean performance, while the shaded area indicates the standard deviation. PROOF **consistently and robustly outperforms other methods by a substantial margin. All methods are based on the same backbone/weight.**

seen and unseen classes. Therefore, when ZS performance is essential, using PROOF$^\dagger$ is the preferred choice.

## D  ADDITIONAL EXPERIMENTAL RESULTS

This section presents further experimental results of PROOF, including comparisons with multiple runs, analysis of parameter numbers, and ablations on projection types. Additionally, we report the results of using OpenAI pre-trained CLIP and provide the full results mentioned in the main paper. We also report the results with larger backbones and collect a new dataset that strictly has no overlapping with the pre-training dataset. We report extensive experiments with this new dataset.

### D.1  MULTIPLE RUNS

Following (Rebuffi et al., 2017), we conduct typical CIL comparisons by randomly splitting the classes with a fixed seed of 1993, and these results are reported in the main paper. In this supplementary section, we perform multiple runs by varying the random seed among {1993, 1994, 1995, 1996, 1997}. We repeat the comparison on CIFAR100 Base50 Inc10 and ImageNet-R Base100 Inc20 five times and present the results in Figure 8. The solid line represents the mean performance, while the shaded area indicates the standard deviation. From these figures, it is evident that PROOF consistently outperforms other methods by a significant margin across different dataset splits. These results validate the robustness of PROOF.

### D.2  PARAMETER ANALYSIS

As mentioned in the main paper, the additional parameters in PROOF come from three sources: the projections, the fusion module, and the visual prototypes. The projection layers are implemented with a single linear layer, each containing $d \times d$ parameters, where $d = 512$ is the embedding dimension. Similarly, the cross-modal fusion is implemented with a single-head self-attention mechanism, and the number of parameters is determined by the weight matrices $W_Q$, $W_K$, and $W_V$, each containing $d \times d$ parameters. The visual prototypes require saving $B \times d$ features, where $B$ is the number of all classes. The number of extra parameters is $(2b + 3) \times d^2 + B \times d$. Hence, these extra parameters are negligible compared to the large backbone of the pre-trained CLIP model, which has approximately 150 million parameters.

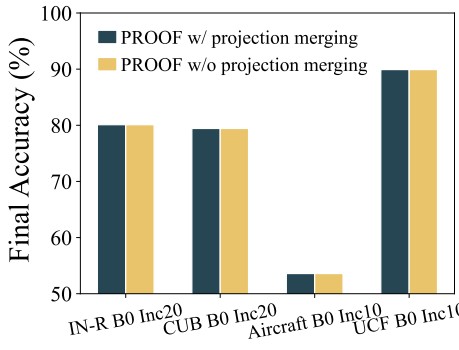
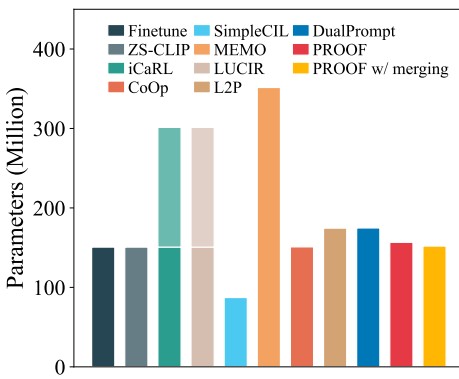

(a) Performance comparison of projection merging        (b) Number of parameters comparison

Figure 9: **Left:** Final performance comparison of PROOF with and without projection merging. Merging the projection modules via Eq. 8 during the testing phase does not hurt the performance but decreases the parameter scale. **Right:** Number of parameters in different methods. The shaded area represents the parameters used during training but dropped during inference. PROOF **achieves state-of-the-art performance with a comparable number of parameters to other methods.**

**Inference Time Merging:** As defined in Eq. 4, the projected embeddings are defined as the summation of all projections. Since these projections are linear layers, we can utilize the associative law of multiplication to merge these projections:

$$P_i(\mathbf{z}) = \sum_{m=1}^{b} P_i^m(\mathbf{z}) = \left( \sum_{m=1}^{b} P_i^m \right)(\mathbf{z}) = \hat{P}_i(\mathbf{z}) \,. \tag{8}$$

As shown in Eq. 8, we can merge all the projections $(P_i^1, P_i^2, \cdots, P_i^b)$ into a single one $(\hat{P}_i)$ using the summation of the weights. Note that $\hat{P}_i$ has the same dimension as the single projection, which means we can alleviate the storage burden of $b$ projections into a single one. This helps us to decrease the extra parameters from $(2b+3) \times d^2 + B \times d$ to $5 \times d^2 + B \times d$. Since $B$ denotes the total number of classes (which ranges from 100 to 300 in current CIL benchmarks), the second term is much smaller than the first term, and the total memory budget is limited by merging all the projections into a single one.

In the implementation, we adopt the projection merging after the last incremental stage and replace the projections with a single one for both visual and textual branches. We show the performance comparison in Figure 9(a) by comparing the model with and without such projection merging. As we can infer from the figure, since these projections are linear ones, merging them via Eq. 8 at the testing stage achieves the same performance as using multiple projections. It must be noted that the merging process can also be done after each incremental stage, while we only conduct it after model training for simplification.

To provide a clear comparison of the parameter numbers for each method, we present the details in Figure 9(b) using CIFAR100 B0 Inc10 as an example. The figure illustrates that PROOF has a similar parameter scale to other finetune-based methods while achieving significantly stronger performance. SimpleCIL, which only utilizes the vision branch, requires fewer parameters for the textual branch but lacks the zero-shot capability. L2P and DualPrompt also only require the vision branch but need an additional encoder to identify the appropriate prompt, resulting in a higher parameter count than PROOF. Additionally, PROOF with projection merging further restricts the number of parameters to be similar to a zero-shot CLIP.

### D.3 Running time comparison

Apart from model size, another important factor for real-world applications is the running time. An ideal continual learning algorithm should perform quickly to efficiently tackle the incoming tasks.

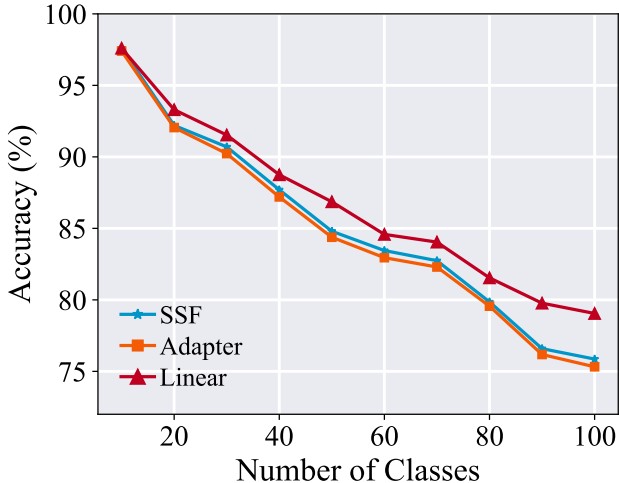

Figure 10: Variations of projection layers. **The choice of using a single linear layer as the projection layer achieves the best performance.**

Table 3: Running time comparison (second) of different methods (**lower is better**). All methods are implemented with a single Tesla V100 GPU.

| Method | CIFAR100 B0 Inc10 | Food B0 Inc10 |
|---|---|---|
| Finetune | 5052 | 8794 |
| CoOp | 7433 | 12997 |
| DualPrompt | 6230 | 10370 |
| L2P | 6742 | 10752 |
| iCaRL | 10678 | 17109 |
| MEMO | 7046 | 11853 |
| DER | 9102 | 15825 |
| PROOF | **4518** | **8697** |

Hence, we measure the running time of different methods in Table 3. We implement all methods with a single Tesla V100 GPU. As we can infer from the table, PROOF has the lowest running time compared to other competitors. It indicates that PROOF has the potential to run efficiently in real-world applications.

### D.4 VARIATION OF PROJECTION TYPES

Apart from simple linear layers, there are other methods to implement the projection layers, such as layer-wise rescale (SSF) (Lian et al., 2022) and Adapter (Houlsby et al., 2019). SSF learns a $d$-dimensional rescale parameter to project the features, while Adapter learns both the down-projection and up-projection for feature mapping. In this section, we explore the performance of these projection methods on CIFAR100 B0 Inc10 and present the results in Figure 10. The figure clearly demonstrates that using a single linear layer as the projection layer achieves the best performance among all methods, indicating its superiority. Furthermore, this result suggests that a simple linear mapping can effectively bridge the gap between visual and textual domains.

### D.5 VARIATION OF CONTEXT INFORMATION

In the main paper, we discuss the composition of the context information **Context**, which should include information from visual prototypes, textual classifiers, and context prompts. In this section, we conduct ablations to demonstrate the effectiveness of constructing **Context** with $[\mathbf{P}, \mathbf{W}, \mathbf{C}]$. Specifically, we perform experiments on CIFAR100 B0 Inc10 and change the context construction

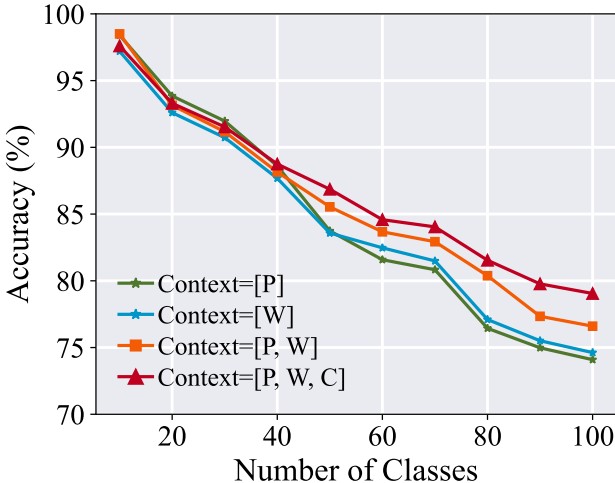

Figure 11: Variations of context information. **The choice of using visual prototypes, textual prototypes, and context prompts as the context information achieves the best performance.**

to **Context** = **P** (visual prototypes only), **Context** = **W** (textual prototypes only), **Context** = [**P**, **W**] (visual and textual prototypes), and **Context** = [**P**, **W**, **C**] (current choice). We keep the same classification rule for these ablations, *i.e.*, classification via Eq. 9. When visual/textual prototypes are not included in the context, we use the projected features without adaptation as the matching target in Eq. 8. The results are presented in Figure 11.

From the results, we observe that using visual prototypes or textual prototypes alone yields similar performance, and the impact of adjustment is marginal. However, when both visual and textual prototypes are jointly utilized as context information, the model can learn from cross-modality and achieve better performance. Lastly, the introduction of context prompts into the context further enhances the performance of PROOF, resulting in the best performance among all variations.

### D.6    DIFFERENT PRE-TRAINED WEIGHTS

In the main paper, we discussed two popular weights for pre-trained CLIP: OpenAI (Radford et al., 2021)[2] and OpenCLIP (Ilharco et al., 2021)[3]. We primarily presented the results of the OpenCLIP pre-trained model in the main paper, while providing the results of the OpenAI weights using a radar chart. In this section, we present the full results of the OpenAI pre-trained CLIP on nine benchmark datasets in Figure 12. The results demonstrate that PROOF consistently achieves the best performance among all methods, regardless of the pre-trained weights used. This highlights the robustness of PROOF in the learning process.

### D.7    LARGER BACKBONES

In the main paper, we mainly compare different methods with CLIP ViT-B/16 backbone and show PROOF outperforms other competitors by a substantial margin. To verify the effectiveness of PROOF with larger backbones, we also conduct experiments with CLIP ViT-L/14 (LAION 400M pre-trained). It is a much larger backbone (427 million parameters) than CLIP ViT-B/16 (149 million parameters) adopted in the main paper. We report the results of different methods in Figure 13.

We can summarize two main conclusions from the figures: 1) Using stronger backbones results in better performance for all compared methods. Since CLIP with ViT-Large contains 427 million parameters, it performs better than ViT-Base. Hence, all of these methods perform better when

---

[2]https://github.com/openai/CLIP
[3]https://github.com/mlfoundations/open_clip

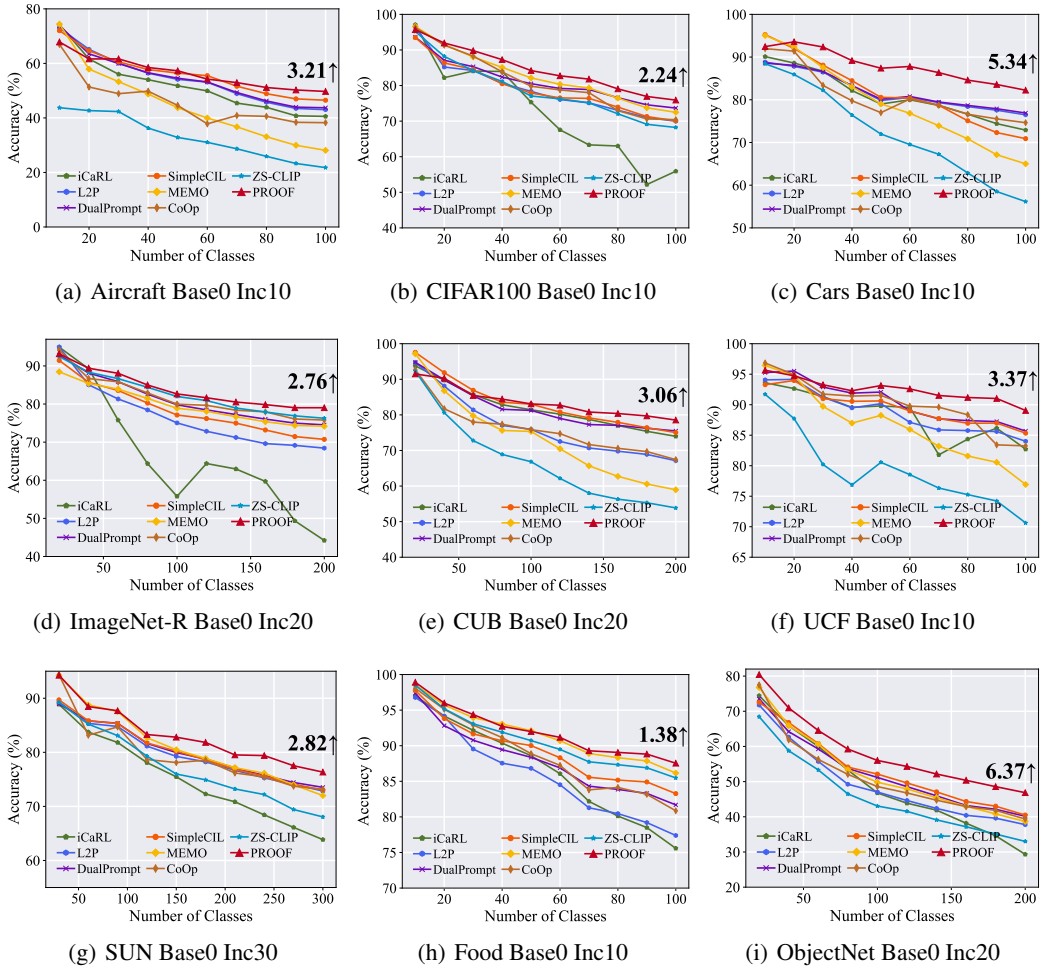

Figure 12: Incremental performance of different methods when using **OpenAI weights**. **All methods are based on the same backbone/weight.** We report the performance gap after the last incremental stage of PROOF and the runner-up method at the end of the line. PROOF **consistently achieves the best performance regardless of the pre-trained weights used.**

changing from ViT-B/16 to ViT-L/14. 2) Given various backbones, PROOF still outperforms these compared methods substantially.

## D.8   EXPERIMENTS WITH IMAGENET

Before the prosperity of pre-trained models, class-incremental learning is mainly evaluated with CIFAR100 (for small-scale evaluation) and ImageNet-100/1000(for large-scale evaluation). Recently, L2P (Wang et al., 2022d) firstly introduces pre-trained models into the class-incremental learning setting. These pre-trained models are often pre-trained with ImageNet-21K, a super-set of ImageNet. Hence, evaluating the incremental learning performance on its subset (*i.e.*, ImageNet 1K) is less meaningful. By contrast, the authors in (Wang et al., 2022d) suggest using datasets that **with large domain gap** to ImageNet for evaluation, *e.g.*, ImageNet-R (Hendrycks et al., 2021).

Since pre-trained CLIP can achieve around 80% zero-shot accuracy on ImageNet, we can assume incremental learning on ImageNet is relatively easy for CLIP. However, we also supply the experimental results on ImageNet100 to make the experiments compatible with former CIL works. We report the experimental results against two typical CIL methods, *i.e.*, FOSTER (Wang et al., 2022a) and DER (Yan et al., 2021) in Figure 14.

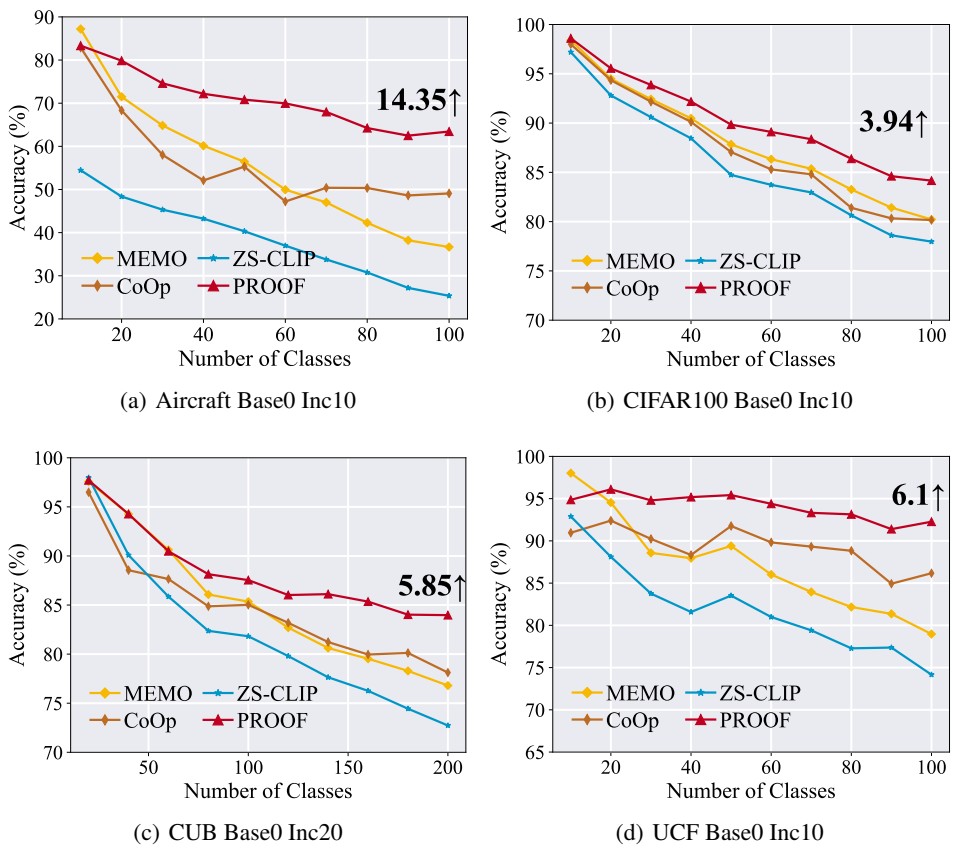

Figure 13: Incremental performance comparison using CLIP with ViT-Large/14 LAION 400M. **All methods are based on the same pre-trained weight and the same backbone.**

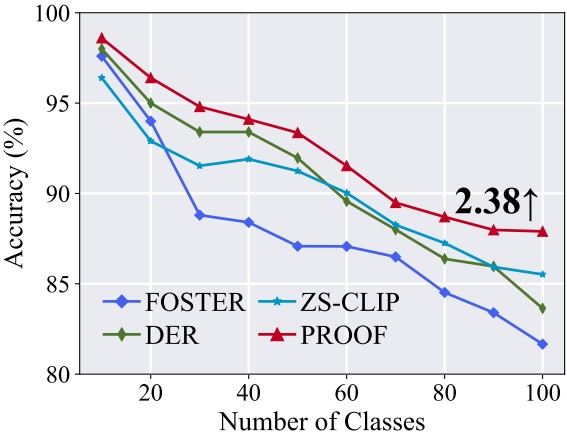

Figure 14: Results on ImageNet100 dataset. PROOF still works competitively on this dataset. All methods use CLIP with ViT-B/16 OpenAI weight for a fair comparison.

We can summarize two main results from the table. First, the performance of zero-shot CLIP is relatively high, indicating that the pre-trained model can already handle the current problem on ImageNet100. Second, our proposed method still outperforms these state-of-the-art methods by a substantial margin, verifying the effectiveness of our method on traditional CIL benchmarks.

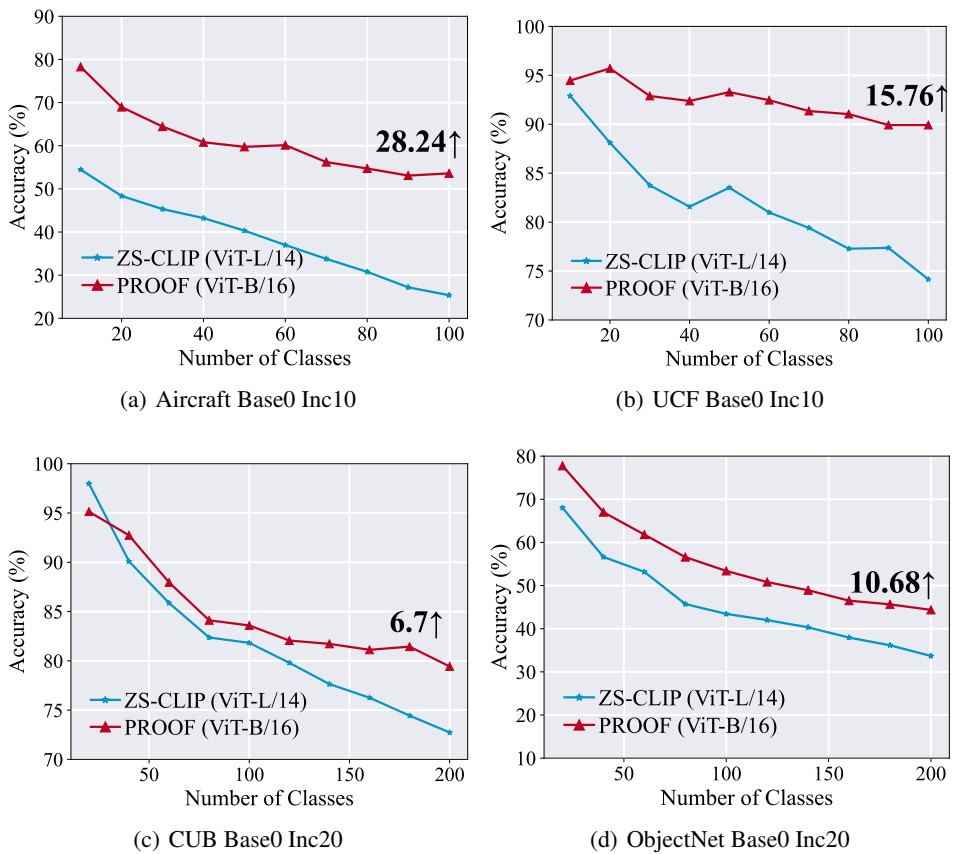

Figure 15: Comparison to zero-shot CLIP with larger backbones. PROOF uses CLIP with ViT-B/16 (156 million), while ZS-CLIP uses CLIP ViT-L/14 (427 million). Both of them are pre-trained with LAION 400M. However, PROOF outperforms ZS-CLIP using only 36% parameters.

## D.9 COMPARISON TO ZERO-SHOT CLIP WITH LARGER BACKBONE

As discussed in Section D.2, PROOF only adds a limited number of parameters (*i.e.*, less than 5%) to a vanilla CLIP. In this section, we compare PROOF to a much larger zero-shot CLIP model to show the effectiveness of these parameters. Specifically, we choose CLIP with ViT-L/14 for the zero-shot CLIP model for comparison, which has 427 million parameters. As we can infer from Figure 15, PROOF beats zero-shot CLIP even only using a much smaller backbone.

## D.10 EXPERIMENTS ON TV100, A STRICTLY NON-OVERLAPPING DATASET TO CLIP

In the main paper, we mainly conduct experiments on **benchmark** datasets that pre-trained CLIP cannot handle, *e.g.*, Aircraft (highly specialized), Food and CUB (fine-grained), and ImageNet-R (out-of-distribution instances). These datasets are popularly adopted as benchmark datasets in tuning pre-trained models (Wang et al., 2022c; Zhou et al., 2022b;a). However, one may still argue that *these pre-trained models may have class overlapping with the pre-training context*. To tackle this problem, we collect a new dataset called **TV100**.

**Dataset Construction:** As we all know, CLIP (Radford et al., 2021) is proposed in ICML 2021, which is trained with image-text pairs (before the year 2021) collected from the Internet. Hence, if we can collect a new dataset after 2021, then we can tell that *CLIP does not know the new knowledge*. To achieve this goal, we select a field with new classes emerging every day, *i.e.*, the TV series.

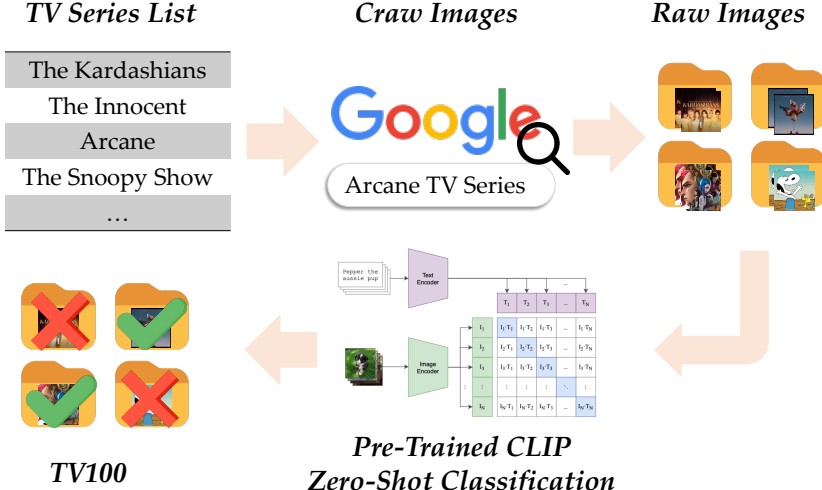

Figure 16: The collection process of TV100. To maintain a non-overlapping knowledge with pre-trained CLIP, we first collect a list of TV series after 2021 (the publication of CLIP). Afterward, we collect data by searching images from Google. However, there may still exist some classes that pre-trained CLIP may know, *e.g.*, "The Kardashians" and "The Snoopy Show". To check whether CLIP knows them, we use a pre-trained CLIP as the filter and delete classes with high zero-shot accuracy. Only hard classes that pre-trained CLIP show limited accuracy are selected in the dataset.

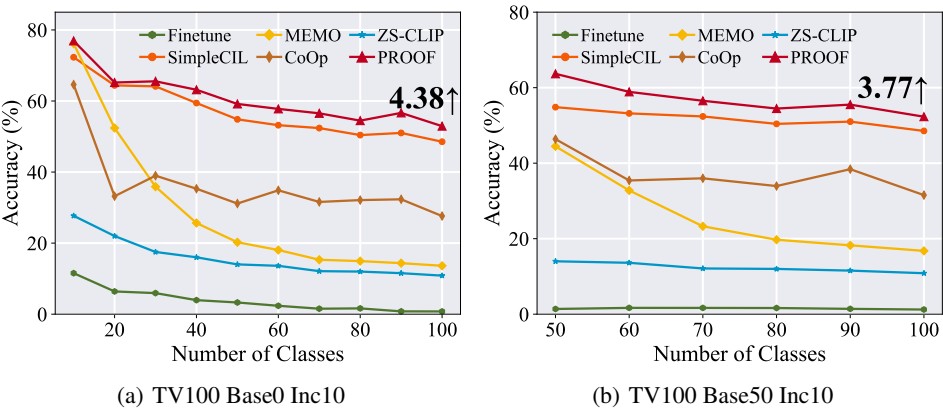

(a) TV100 Base0 Inc10                    (b) TV100 Base50 Inc10

Figure 17: Experiments on TV100. **All methods are based on the same pre-trained weight (CLIP ViT-B/16 LAION400M) and the same backbone.** TV100 is a non-overlapping dataset containing images of TV series after 2021 (the release of CLIP), which properly evaluates the ability of different continual learning algorithms to learn new knowledge. PROOF outperforms other compared methods by a substantial margin.

Specifically, we manually search for TV series from IMDB and collect the items released after 2021[4]. Afterward, we download the related images on Google by searching the keyword "**[NAME] TV Series**," where [NAME] is the name of the TV series. The downloaded images are then processed manually to delete repeated and meaningless ones. Hence, we can get a large dataset that contains around 800 classes.

---

[4]For those series with multiple seasons, we directly drop them since CLIP may have seen a former season, *e.g.*, Stranger Things Season 4 is released in 2022, while Stranger Things Seasons 1, 2, and 3 are released before 2021.

However, some of these classes may not be "new" for a pre-trained CLIP, *e.g.*, "The Kardashians" was released in 2022 while it is not a new concept for CLIP because the Kardashian–Jenner family has been popular in America since the last century. A similar phenomenon also occurs in "The Snoopy Show" (Snoopy is a famous cartoon character) and "The Cuphead Show" (Cuphead is a video game released in 2017). Hence, **we need to select some challenging classes that CLIP does not know from the TV series pool.** Correspondingly, we use a pre-trained CLIP to rank the difficulty of these classes by measuring the zero-shot accuracy of each image and the text "a photo of the TV series [CLASS]." We choose the top-100 hard classes based on the zero-shot accuracy and construct the TV100 dataset. The collection process is summarized in Figure 16. Surprisingly, a pre-trained CLIP only achieves around 10% accuracy on this dataset, verifying that CLIP does not master these classes. Besides, since the dataset is collected after the publication of CLIP, there is no class overlapping between pre-trained CLIP and TV100. **We will make this dataset publicly available upon acceptance.**

Correspondingly, we also conduct experiments on this new dataset. With the other settings the same as the main paper, we select two dataset splits (*i.e.*, Base0 Inc10 and Base50 Inc10) and report the results in Figure 17. We can summarize two main conclusions from the figure. Firstly, zero-shot CLIP performs poorly on this dataset, verifying that this dataset perfectly serves as the benchmark to evaluate the continual learning ability of pre-trained CLIP. Secondly, PROOF still outperforms other competitors by a substantial margin, verifying its strong performance in real-world continual learning tasks.

### D.11 FULL RESULTS

We provide the complete results of the benchmark comparison in the main paper, which are presented in Table 4 and Figures 18 and 19. These results are obtained using OpenCLIP pre-trained weights on LAION-400M (Ilharco et al., 2021). Table 4 displays the average and last accuracy for the nine benchmark datasets. Figures 18 and 19 illustrate the incremental performance with varying numbers of base classes. Across all these evaluations, PROOF consistently outperforms the compared methods, demonstrating its superior performance.

## E EXPERIMENTAL DETAILS

This section provides detailed information about the experiments conducted, including the introduction of datasets, exemplar selection, and the methods compared in the paper.

### E.1 DISCUSSIONS ABOUT DATASET SELECTION

In the main paper, we evaluate different CIL algorithms on nine datasets, which are selected based on the following criteria:

- **CIL Benchmark:** We follow the benchmark pre-trained model-based CIL method (Wang et al., 2022c), where **data with a large domain gap to the pre-train model** is used for continual learning. For example, ImageNet-R contains art, cartoon, and sketch-style images that are out-of-the distribution of the pre-trained model. Similarly, ObjectNet contains objects from new viewpoints on new backgrounds that are hard for the pre-trained model. Given the domain gap, directly applying zero-shot learning on these datasets performs poorly.

- **VLM-Tuning Benchmark:** We also follow the typical VLM-tuning benchmark (Zhou et al., 2022a;b) to use **specialized or fine-grained datasets that pre-trained VLM cannot handle**. For example, pre-trained CLIP cannot differentiate a "Boeing 707" from a "Boeing 747", while our proposed method improves its performance from 20% to 60%. Similar cases also include the classification between "Audi 100 Sedan 1994" and "Audi 100 Wagon 1994", where our method boosts the performance by 15%.

- **Non-overlapping Data:** Apart from the above nine benchmarks from CIL and VLM tuning, we have also collected a new dataset (*i.e.*, TV100 in Section D.10) that has no data overlapping to pre-trained CLIP. This dataset's collection is strictly after CLIP's publication, which is clearly "new" and proper to serve as the dataset for pre-trained CLIP.

Table 4: Average and last performance comparison of different methods. The first and second columns represent the methods with and without exemplars. The performance of L2P and DualPrompt are reproduced with the source code with exemplars. The best performance is shown in bold. **All methods are based on the same backbone/weight.**

| Method | Exemplar | Aircraft | | | | CIFAR100 | | | | Cars | | | |
| | | B0 Inc10 | | B50 Inc10 | | B0 Inc10 | | B50 Inc10 | | B0 Inc10 | | B50 Inc10 | |
| | | $\bar{A}$ | $A_B$ | $\bar{A}$ | $A_B$ | $\bar{A}$ | $A_B$ | $\bar{A}$ | $A_B$ | $\bar{A}$ | $A_B$ | $\bar{A}$ | $A_B$ |
|---|---|---|---|---|---|---|---|---|---|---|---|---|---|
| Finetune | ✗ | 3.16 | 0.96 | 1.72 | 1.05 | 7.84 | 4.44 | 5.30 | 2.46 | 3.14 | 1.10 | 1.54 | 1.13 |
| Finetune LiT (Zhai et al., 2022) | ✗ | 27.74 | 14.28 | 25.10 | 13.77 | 44.66 | 14.69 | 27.69 | 7.67 | 84.12 | 72.37 | 83.08 | 78.23 |
| Finetune CoOp (Zhou et al., 2022b) | ✗ | 14.54 | 7.14 | 13.05 | 7.77 | 47.00 | 24.24 | 41.23 | 24.12 | 36.46 | 21.65 | 37.40 | 20.87 |
| SimpleCIL (Zhou et al., 2023c) | ✗ | 59.24 | 48.09 | 53.05 | 48.09 | 84.15 | 76.63 | 80.20 | 76.63 | 92.04 | 86.85 | 88.96 | 86.85 |
| ZS-CLIP (Radford et al., 2021) | ✗ | 26.66 | 17.22 | 21.70 | 17.22 | 81.81 | 71.38 | 76.49 | 71.38 | 82.60 | 76.37 | 78.32 | 76.37 |
| CoOp (Zhou et al., 2022b) | ✓ | 44.26 | 39.87 | 41.81 | 39.18 | 83.37 | 73.36 | 78.34 | 73.04 | 89.73 | 84.91 | 87.98 | 86.60 |
| iCaRL (Rebuffi et al., 2017) | ✓ | 53.60 | 43.98 | 50.40 | 45.33 | 79.91 | 63.94 | 71.94 | 63.00 | 89.38 | 84.95 | 86.71 | 84.19 |
| MEMO (Zhou et al., 2023b) | ✓ | 42.24 | 25.41 | 38.16 | 27.75 | 84.67 | 74.98 | 80.75 | 75.34 | 88.23 | 81.31 | 84.90 | 81.83 |
| L2P (Wang et al., 2022d) | ✓ | 55.06 | 44.88 | 47.78 | 43.37 | 76.42 | 66.21 | 72.67 | 67.88 | 83.81 | 72.44 | 79.76 | 73.47 |
| DualPrompt (Wang et al., 2022c) | ✓ | 55.95 | 46.53 | 50.93 | 46.50 | 79.07 | 70.06 | 74.81 | 70.75 | 85.30 | 74.35 | 81.32 | 75.85 |
| PROOF | ✓ | **61.00** | **53.59** | **59.99** | **58.90** | **86.70** | **79.05** | **82.92** | **78.87** | **93.26** | **89.84** | **90.53** | **89.54** |

| Method | Exemplar | ImageNet-R | | | | CUB | | | | UCF | | | |
| | | B0 Inc20 | | B100 Inc20 | | B0 Inc20 | | B100 Inc20 | | B0 Inc10 | | B50 Inc10 | |
| | | $\bar{A}$ | $A_B$ | $\bar{A}$ | $A_B$ | $\bar{A}$ | $A_B$ | $\bar{A}$ | $A_B$ | $\bar{A}$ | $A_B$ | $\bar{A}$ | $A_B$ |
|---|---|---|---|---|---|---|---|---|---|---|---|---|---|
| Finetune | ✗ | 1.37 | 0.43 | 1.01 | 0.88 | 2.06 | 0.64 | 0.56 | 0.47 | 4.51 | 1.59 | 1.21 | 0.80 |
| Finetune LiT (Zhai et al., 2022) | ✗ | 64.88 | 30.42 | 57.75 | 29.77 | 58.15 | 35.28 | 51.95 | 35.96 | 79.25 | 64.84 | 81.79 | 65.40 |
| Finetune CoOp (Zhou et al., 2022b) | ✗ | 60.73 | 37.52 | 54.20 | 39.77 | 27.61 | 8.57 | 24.03 | 10.14 | 47.85 | 33.46 | 42.02 | 24.74 |
| SimpleCIL (Zhou et al., 2023c) | ✗ | 81.06 | 74.48 | 76.84 | 74.48 | 83.81 | 77.52 | 79.75 | 77.52 | 90.44 | 85.68 | 88.12 | 85.68 |
| ZS-CLIP (Radford et al., 2021) | ✗ | 83.37 | 77.17 | 79.57 | 77.17 | 74.38 | 63.06 | 67.96 | 63.06 | 75.50 | 67.64 | 71.44 | 67.64 |
| CoOp (Zhou et al., 2022b) | ✓ | 82.40 | 76.20 | 79.76 | 77.13 | 77.34 | 68.70 | 74.09 | 67.47 | 90.13 | 86.24 | 88.36 | 85.71 |
| iCaRL (Rebuffi et al., 2017) | ✓ | 72.22 | 54.38 | 68.67 | 60.15 | 82.04 | 74.74 | 78.57 | 75.07 | 89.47 | 84.34 | 88.51 | 84.11 |
| MEMO (Zhou et al., 2023b) | ✓ | 80.00 | 74.07 | 76.72 | 73.95 | 77.32 | 65.69 | 72.88 | 66.41 | 84.02 | 74.08 | 82.58 | 75.48 |
| L2P (Wang et al., 2022d) | ✓ | 75.73 | 67.22 | 74.15 | 71.20 | 79.23 | 68.54 | 75.85 | 71.12 | 88.71 | 83.93 | 86.51 | 83.22 |
| DualPrompt (Wang et al., 2022c) | ✓ | 78.47 | 70.82 | 72.98 | 69.18 | 83.21 | 74.06 | 78.06 | 74.27 | 89.48 | 85.41 | 86.96 | 84.65 |
| PROOF | ✓ | **85.34** | **80.10** | **82.32** | **80.30** | **84.93** | **79.43** | **81.67** | **79.18** | **92.34** | **89.92** | **91.70** | **89.16** |

| Method | Exemplar | SUN | | | | Food | | | | ObjectNet | | | |
| | | B0 Inc30 | | B150 Inc30 | | B0 Inc10 | | B50 Inc10 | | B0 Inc20 | | B100 Inc20 | |
| | | $\bar{A}$ | $A_B$ | $\bar{A}$ | $A_B$ | $\bar{A}$ | $A_B$ | $\bar{A}$ | $A_B$ | $\bar{A}$ | $A_B$ | $\bar{A}$ | $A_B$ |
|---|---|---|---|---|---|---|---|---|---|---|---|---|---|
| Finetune | ✗ | 4.51 | 1.59 | 0.78 | 0.72 | 3.49 | 1.71 | 2.14 | 1.52 | 1.34 | 0.47 | 0.69 | 0.54 |
| Finetune LiT (Zhai et al., 2022) | ✗ | 79.25 | 64.84 | 38.23 | 20.00 | 40.62 | 12.96 | 29.74 | 12.05 | 43.27 | 17.46 | 32.85 | 17.17 |
| Finetune CoOp (Zhou et al., 2022b) | ✗ | 45.93 | 23.11 | 39.33 | 24.89 | 36.01 | 14.18 | 33.13 | 18.67 | 21.24 | 6.29 | 16.21 | 6.82 |
| SimpleCIL (Zhou et al., 2023c) | ✗ | 82.13 | 75.58 | 78.62 | 75.58 | 87.89 | 81.65 | 84.73 | 81.65 | 52.06 | 40.13 | 45.11 | 40.13 |
| ZS-CLIP (Radford et al., 2021) | ✗ | 79.42 | 72.11 | 74.95 | 72.11 | 87.86 | 81.92 | 84.75 | 81.92 | 38.43 | 26.43 | 31.12 | 26.43 |
| CoOp (Zhou et al., 2022b) | ✓ | 80.46 | 73.44 | 77.68 | 73.06 | 85.38 | 76.15 | 81.74 | 76.35 | 46.16 | 33.81 | 40.40 | 34.47 |
| iCaRL (Rebuffi et al., 2017) | ✓ | 78.56 | 67.30 | 74.74 | 69.07 | 84.12 | 71.68 | 78.86 | 70.64 | 45.28 | 26.97 | 37.22 | 26.15 |
| MEMO (Zhou et al., 2023b) | ✓ | 81.48 | 73.45 | 78.00 | 73.87 | 89.18 | 82.85 | 86.50 | 83.08 | 46.98 | 33.37 | 41.62 | 34.67 |
| L2P (Wang et al., 2022d) | ✓ | 79.83 | 72.14 | 76.16 | 72.32 | 84.48 | 75.22 | 85.04 | 80.56 | 46.18 | 34.00 | 43.90 | 39.57 |
| DualPrompt (Wang et al., 2022c) | ✓ | 80.14 | 73.06 | 77.25 | 73.82 | 87.12 | 81.27 | 85.37 | 82.36 | 53.13 | 40.59 | 45.84 | 40.37 |
| PROOF | ✓ | **83.57** | **77.28** | **80.70** | **77.49** | **90.04** | **84.73** | **87.52** | **84.74** | **55.28** | **44.36** | **49.64** | **43.65** |

In summary, the datasets adopted in this paper are based on the popularly acknowledged benchmark (Zhou et al., 2022a;b; Wang et al., 2022c). These datasets are known to have a large domain gap to the pre-trained data or highly specialized and fine-grained. Directly conducting zero-shot learning works poorly or even fails on these datasets. Hence, they are proper to be adopted to evaluate the continual learning algorithm of a pre-trained model. Additionally, we also evaluate the performance of different methods on a dataset with no data overlapping to the pre-trained CLIP. Extensive experiments on these ten datasets verify the effectiveness of the proposed method.

## E.2 DATASET INTRODUCTION

In our evaluation, we utilize nine datasets, which are introduced in Table 5 in the main paper. It is worth noting that some of these datasets have a larger number of classes, but we select a subset of classes for ease of data split and evaluation.

**Exemplar Selection:** As mentioned in the main paper, we follow the exemplar selection approach in (Rebuffi et al., 2017; Wu et al., 2019; Hou et al., 2019) to utilize herding algorithm (Welling, 2009). In addition, there are two typical methods (Zhou et al., 2023a) to store these exemplars in memory.

1. *Fixed Memory Budget:* In this approach, a fixed memory budget of $K$ instances is allocated. Given the number of seen classes denoted as $|\mathcal{Y}_b|$, the model selects $\frac{K}{|\mathcal{Y}_b|}$ exemplars per class after each incremental stage.

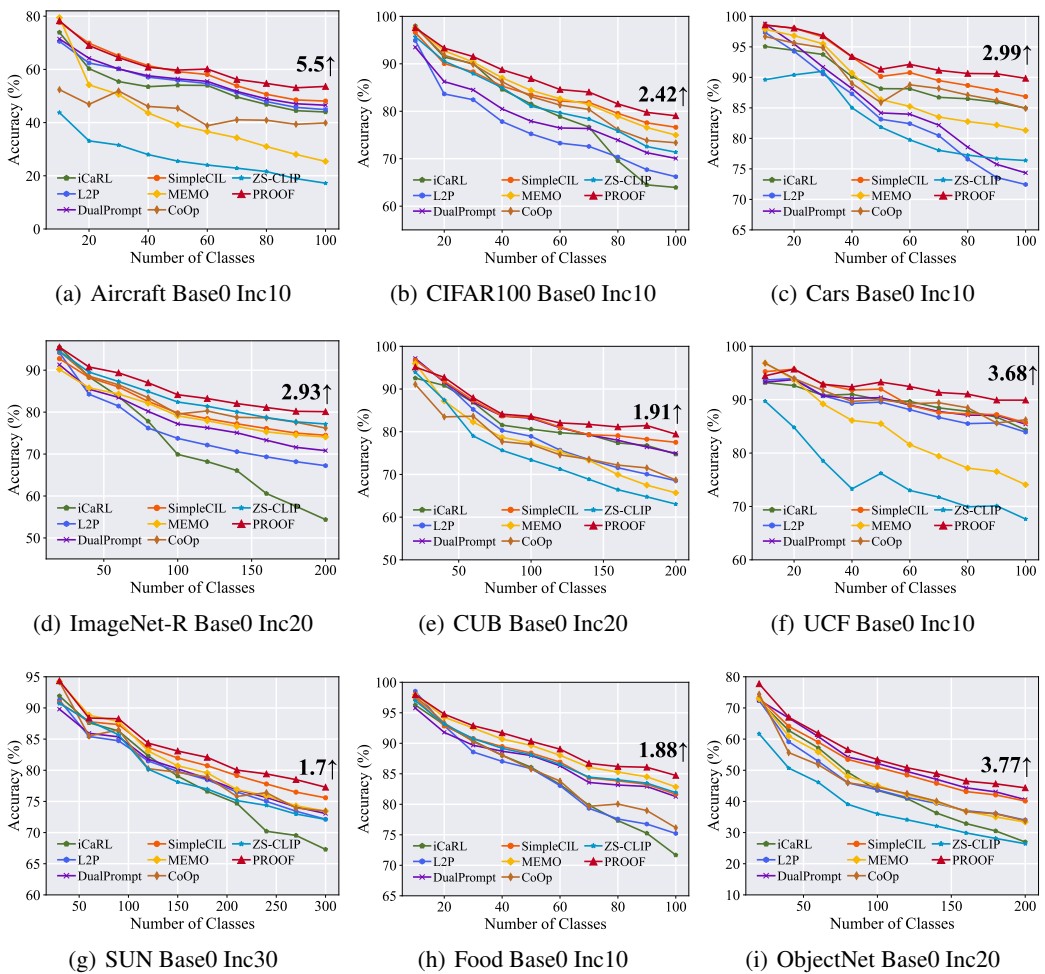

Figure 18: Incremental performance of different methods. We report the performance gap after the last incremental stage of PROOF and the runner-up method at the end of the line. **All methods are based on the same backbone/weight.**

2. *Expandable Exemplar Set:* In this method, an expandable exemplar set is maintained as the data evolves. With the number of exemplars per class denoted as $k$, the model stores $|\mathcal{Y}_b| \times k$ exemplars in total after each incremental stage.

We evaluate both protocols using these benchmark datasets in our experiments. Specifically, we employ the first policy for CIFAR100 and Food, keeping a total of 2,000 exemplars. Since these datasets consist of 100 classes, the average number of exemplars per class after the last incremental stage is 20. We adopt the second policy for the other datasets and store 20 exemplars per class.

### E.3 COMPARED METHODS INTRODUCTION AND COMPARISON FAIRNESS

This section provides an overview of the compared methods discussed in the main paper. **Please note that all compared methods are based on the same pre-trained CLIP model. In other words, we have adapted these original algorithms with pre-trained CLIP for a fair comparison.** These methods, listed in the order presented in Table 4, include:

- **Finetune:** This baseline method involves finetuning the pre-trained CLIP model using contrastive loss. No regularization terms are set, and no part of the model is frozen, allowing us to observe the forgetting phenomenon in sequential learning.

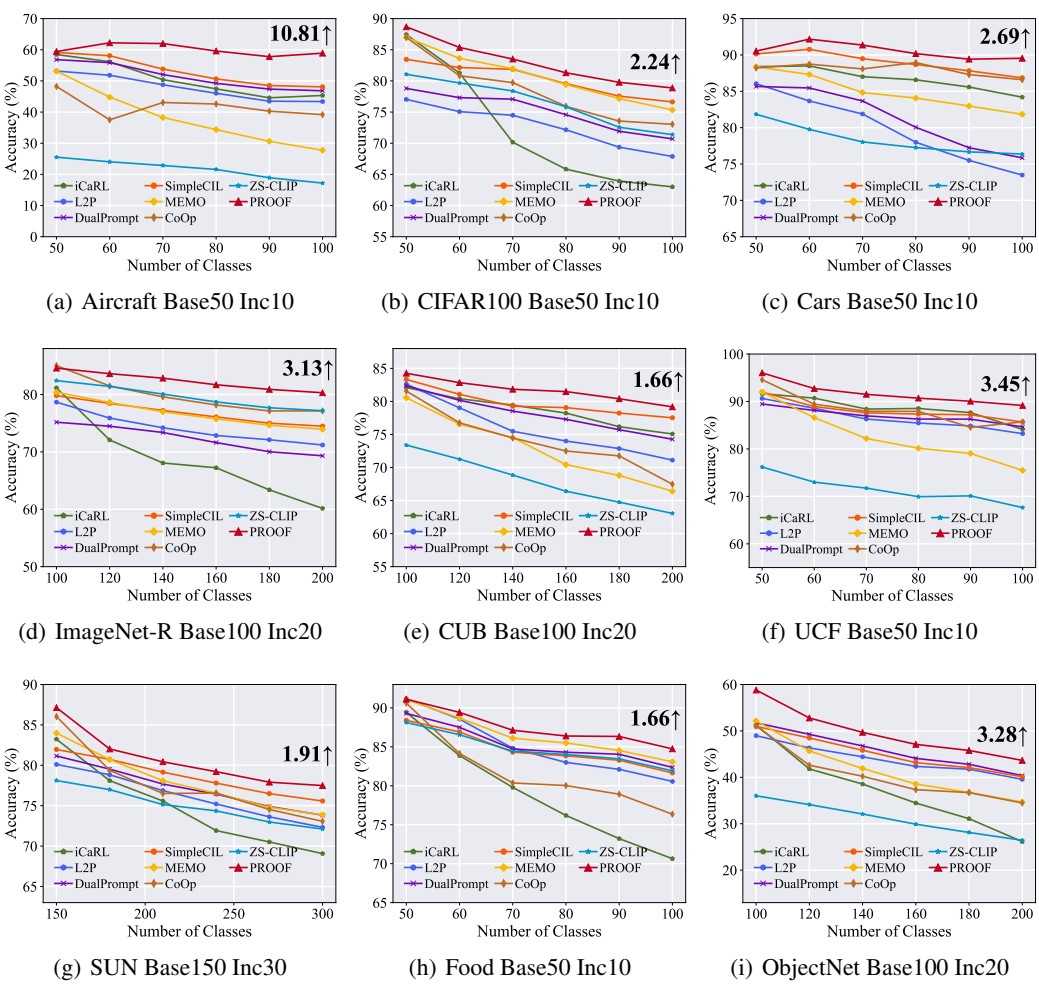

Figure 19: Incremental performance of different methods with large base classes. We report the performance gap after the last incremental stage of PROOF and the runner-up method at the end of the line. **All methods are based on the same backbone/weight.**

Table 5: Introduction about benchmark datasets.

| Dataset | # training instances | # testing instances | # Classes | Link |
|---|---|---|---|---|
| CIFAR100 | 50,000 | 10,000 | 100 | Link |
| CUB200 | 9,430 | 2,358 | 200 | Link |
| ImageNet-R | 24,000 | 6,000 | 200 | Link |
| ObjectNet | 26,509 | 6,628 | 200 | Link |
| Aircraft | 6,667 | 3,333 | 100 | Link |
| Cars | 4,135 | 4,083 | 100 | Link |
| UCF | 10,053 | 2,639 | 100 | Link |
| SUN | 72,870 | 18,179 | 300 | Link |
| Food | 79,998 | 20,012 | 100 | Link |

- **Finetune LiT (Zhai et al., 2022):** Following LiT, which freezes the image encoder and only finetunes the text encoder, we design Finetune LiT with CIL. Similar to finetune, we sequentially tune the pre-trained CLIP with contrastive loss while the image encoder is frozen during optimization.

- **Finetune CoOp (Zhou et al., 2022b):** Following the CoOp method, this approach freezes both the image encoder and text encoder of the pre-trained CLIP. It optimizes a learnable prompt tensor $\mathbf{t}$ (as in Eq.4) using contrastive loss without utilizing any historical data for rehearsal.

- **SimpleCIL (Zhou et al., 2023c):** This method relies on the pre-trained image encoder and does not involve the text encoder. Hence, in the pre-trained CLIP, we drop the text branch and only use the visual branch for evaluation. The frozen image encoder extracts class centers (prototypes) for each new class, and a cosine classifier is utilized for classification. Since the model is not updated via backpropagation, it showcases the generalizability of the pre-trained vision encoder on downstream tasks.

- **ZS-CLIP (Radford et al., 2021):** This baseline freezes the pre-trained CLIP and predicts the logits of each incoming class using cosine similarity (Eq. 2). It serves as a reference for the performance of pre-trained CLIP on downstream tasks.

- **CoOp (with exemplars):** This method combines the CoOp approach with exemplar rehearsal. During learning new classes, the model utilizes a combination of the current dataset and exemplar set to optimize the learnable prompt.

- **iCaRL (Rebuffi et al., 2017):** iCaRL is a typical class-incremental learning algorithm that employs knowledge distillation and exemplar replay to mitigate forgetting. To make it compatible with the CLIP backbone, we combine the contrastive loss with distillation loss to learn new classes while retaining knowledge of old classes. The distillation loss is built between the predicted logits of the old and new model in order to reflect the old model's behavior in the updated model.

- **LUCIR (Hou et al., 2019):** is a typical class-incremental learning algorithm that combines feature distillation and metric learning. To make it compatible with CLIP's structure, we apply feature distillation for both visual and textual branches in order to resist forgetting former knowledge.

- **DER (Yan et al., 2021):** a state-of-the-art class-incremental learning algorithm. When facing a new task, it creates a new backbone and concatenates the features of the old and new backbone to train a new FC layer. The memory budget will linearly increase as data evolves since it keeps $B$ backbones in memory. To make it compatible with pre-trained CLIP, we expand both the visual and textual branch for a new task. During inference, we concatenate all the visual branches for the image embedding and concatenate all the textual branches for the textual information. The inference function is still the same with Eq. 2 by matching the visual-textual embeddings.

- **MEMO (Zhou et al., 2023b):** MEMO extends DER by decoupling the backbone into generalized and specialized blocks and only expanding specialized blocks for new tasks based on the shared generalized blocks. As a state-of-the-art class-incremental learning algorithm based on network expansion, MEMO is modified to be compatible with the pre-trained CLIP. The image and text encoders are expanded for new tasks, and the concatenated features are used for prediction based on cosine similarity. In the implementation, we treat the last transformer block of the visual and textual branches as generalized blocks and expand them for each new task.

- **L2P (Wang et al., 2022d):** L2P is a state-of-the-art class-incremental learning algorithm utilizing pre-trained vision transformers. In this case, the text encoder of pre-trained CLIP is dropped, and a prompt pool is learned to adapt to evolving data. Another pre-trained image encoder is required to select the appropriate prompt during inference.

- **DualPrompt (Wang et al., 2022c):** DualPrompt is an extension of L2P that incorporates two types of prompts: general and expert prompts. It also relies on another pre-trained image encoder for prompt retrieval.

**Comparison Fairness:** It is clear that **all these methods are compared fairly, *i.e.*, initialized with the same pre-trained weights and using the same number of exemplars for incremental learning**. Although some algorithms are proposed for CNN, the basic idea is still compatible with the CLIP structure, and we modify their backbone into the pre-trained CLIP for a fair comparison among different algorithms.

Table 6: Forgetting measure of different methods (**lower is better**). All methods use the same backbone and same number of exemplars.

| Method | Aircraft B0 Inc10 | Cars B0 Inc10 | CIFAR100 B0 Inc10 | UCF B0 Inc10 |
|--------|-------------------|---------------|-------------------|--------------|
| CoOp   | 28.20             | 7.43          | 20.74             | 10.81        |
| iCaRL  | 13.44             | 4.18          | 31.45             | 6.66         |
| MEMO   | 17.34             | 4.77          | 12.40             | 9.78         |
| PROOF  | **9.41**          | **3.51**      | **8.02**          | **4.07**     |

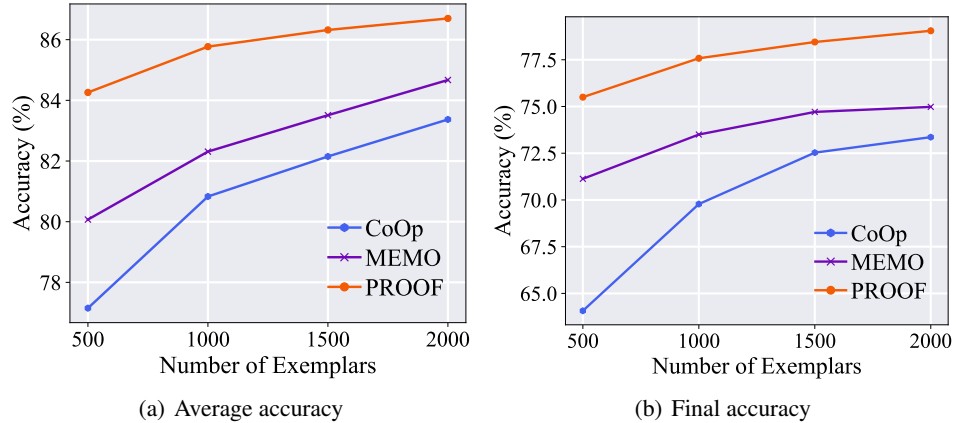

(a) Average accuracy      (b) Final accuracy

Figure 20: Accuracy trend with different number of exemplars.

In summary, all the compared methods are initialized with the same backbone (*i.e.*, pre-trained CLIP with the same initial weight), with no exception. **Hence, experimental results perfectly and fairly reflect the continual learning ability of different algorithms with new classes.**

### E.4 FORGETTING MEASURE

Apart from the commonly adopted measure $\mathcal{A}_B$ and $\bar{\mathcal{A}}$, we can also measure the forgetting degree of different methods using the Forgetting metric (Wang et al., 2023a; Mai et al., 2022). Specifically, we use $a_{k,j}$ to denote the performance of task $j$ after learning task $k$. The forgetting measure after learning task $k$ is defined as:

$$\text{Forgetting}_k = \frac{1}{k-1} \sum_{j=1}^{k-1} f_{j,k}, \tag{9}$$

where $f_{j,k}$ is defined as:

$$f_{j,k} = \max_{i \in \{1,\dots,k-1\}} (a_{i,j} - a_{k,j}), \forall j < k. \tag{10}$$

Hence, Forgetting measures the gap between the best performance and the final performance, and a lower forgetting measure denotes better performance in resist forgetting. We report Forgetting$_B$ of different methods in Table 6 for a comparison.

As we can infer from the table, PROOF shows the lowest forgetting among all competitors, indicating its best performance to resist catastrophic forgetting.

### E.5 PERFORMANCE CHANGE WITH EXEMPLAR NUMBER

In the main paper, we mainly follow the benchmark setting (Rebuffi et al., 2017) to utilize 2000 exemplars in total. In this section, we conduct experiments by changing the number of exemplars to show the performance trend. We conduct experiences on CIFAR100 B0 Inc10, and vary the number

of exemplars among $\{500, 1000, 1500, 2000\}$. We report the average accuracy and final accuracy of PROOF, CoOp, and MEMO in Figure 20.

As we can infer from the figure, there are two main conclusions. 1) All methods facilitate from more exemplars, and they achieve better performance as exemplar increases. 2) However, PROOF consistently achieves the best performance among all methods at different exemplar scales. Experiments verify the effectiveness of PROOF in various settings.

### E.6    DISCUSSIONS OF RELATED WORKS

In this paper, we design PROjectiOn Fusion (**PROOF**) that enables vision-language models to learn without forgetting. Given a pre-trained vision-language model, we freeze the visual and textual embedding functions and learn expandable projections upon them. During the optimization process, we encode task-specific information into these projection layers to extract more adaptive features of downstream tasks. On the other hand, we also freeze prior projections when learning new ones, which enables the latest projection to learn the most effective residual features to capture all seen classes. Hence, catastrophic forgetting can be alleviated during the projection expansion process.

Different from the "feature projection" in this paper, there are some works (Deng et al., 2021; Saha et al., 2021; Lin et al., 2022) addressing "gradient projection" or "parameter projection" in the continual learning field. Among them, (Saha et al., 2021) suggests taking gradient steps in the orthogonal direction to the gradient subspaces deemed important for the past tasks. Based on this, (Deng et al., 2021) introduces a soft weight to represent the importance of each basis representing past tasks in gradient projection memory. Similarly, (Lin et al., 2022) introduces a notion of 'trust region' to select the most related old tasks for the new task in a layer-wise and single-shot manner, using the norm of gradient projection onto the subspace spanned by task inputs. **As we can infer from these works, although both talk about "projection", the ideas and operations between** PROOF **and these works are essentially different.** Specifically, these works (Deng et al., 2021; Saha et al., 2021; Lin et al., 2022) aim to project gradient updating directions not to harm existing knowledge. By contrast, PROOF works on feature projection, which projects the pre-trained features into parallel subspaces to build task-specific features. Hence, they differ substantially from the motivation to the operation, except for the similar name of "projection."

Apart from these differences, we highlight the contributions of PROOF in this section. Through expandable task-specific projections, we enable a pre-trained vision-language model for continual learning. We also design a cross-modal fusion module to contextualize the embeddings and less forgetting. The ideas of learning task-specific projections, learning cross-modal fusion with context information, and joint adaptation with context prompts are innovative and have not been explored by any prior work. PROOF is a unified framework that can be applied to various continual learning scenarios (*e.g.*, class-incremental learning and continual cross-modal retrieval) and various vision-language models (*e.g.*, CLIP and BEiT-3), showing strong performance in various applicative scenarios.

## F    BROADER IMPACTS

In this work, we address the class-incremental learning problem with vision-language models, which is a fundamental challenge in machine learning. Our focus is on tackling the forgetting problem that arises when sequentially finetuning a vision-language model. We propose solutions to project and integrate features from multiple modalities for unified classification. Our research provides valuable insights for applications that struggle with managing the forgetting issue in large pre-trained vision-language models. However, there are still ample opportunities for further exploration in this field. Therefore, we aspire to stimulate discussions on class-incremental learning in real-world scenarios and encourage more research to develop practical models for this purpose.

We also acknowledge the ethical considerations associated with this technology. It is crucial to recognize that individuals expect learning systems to refrain from storing any personal information for future rehearsal. While there are risks involved in AI research of this nature, we believe that developing and demonstrating such techniques are vital for comprehending both the beneficial and potentially concerning applications of this technology. Our aim is to foster discussions regarding best practices and controls surrounding these methods, promoting responsible and ethical utilization of technology.

