# OpenReview forum: "Learning without Forgetting for Vision-Language Models"
_ICLR.cc/2024/Conference — Submitted to ICLR 2024_

### Official Review · Reviewer_s6fA · 2023-10-30

**Soundness:** 3 good
**Presentation:** 2 fair
**Contribution:** 2 fair
**Rating:** 5
**Confidence:** 4

**Summary:**

The paper proposes one of the first mechanism to do continual learning with Vision-Language Models (VLM) such as CLIP. Through a system of projectors and a revised definition of context, the authors tested their model, PROOF, on a variety of datasets for continual learning obtaining state-of-the-art performances.

**Strengths:**

- The authors tested for the first time a VLM model for continual learning.
- The authors tested their PROOF on a variety of datasets testing the effectiveness of the model.
- The authors proved the effectiveness of the model with very interesting and detailed ablation studies.

**Weaknesses:**

- The paper lacks motivation and innovation: The authors suggest using CLIP for class-incremental continual learning, but it would be more interesting to see its performance on tasks like incremental captioning or retrieval. Unlike L2P, where a large pretrained model was used, CIL could have been just one application.
- Furthermore, the PROOF mechanism, while innovative, lacks depth. Projection networks are common in continual learning, and the new context definition isn't explored.
- The main paper lacks standard deviation in results and doesn't consider multiple runs with different class orders.
- There's no analysis of time and memory usage, except for a basic mention of memory requirements in supplementary materials.
- The paper's narration could also be improved

**Questions:**

- It looks like the supplementary materials are more informative and present more interesting results w.r.t. the main paper. Why did the authors exclude them from the main paper?
- The definition of W is not reported in the paper. How W is defined in the context?
- Can the authors provide an analysis of the accuracies of the model varying the number of exemplars?

---

> ### Author Response · Authors · 2023-11-19
> **Response to Reviewer s6fA (1/3)**
>
> We thank the reviewer for the feedback and comments. We have carefully revised the manuscript following the reviewer’s comments, and the equation/section numbers in the rebuttal refer to the latest version. We respond to the concerns below:
>
> **Q1** The paper lacks motivation and innovation: The authors suggest using CLIP for class-incremental continual learning, but it would be more interesting to see its performance on tasks like incremental captioning or retrieval. Unlike L2P, where a large pretrained model was used, CIL could have been just one application.
>
> **A1** We thank the reviewer for the suggestion. Just as the reviewer pointed out, we not only investigate PROOF’s strong performance in class-incremental learning in this paper. Additionally, **we have compared PROOF to other competitors in the continual cross-modal retrieval tasks in Section 5.4 and Figure 4 (a) in the initial submission**. Correspondingly, we find PROOF is a concise and unified framework that solves catastrophic forgetting in various continual learning scenarios (e.g., class-incremental learning and continual cross-modal retrieval).
>
> **Q2** Furthermore, the PROOF mechanism, while innovative, lacks depth. Projection networks are common in continual learning, and the new context definition isn't explored.
>
> **A2** We thank the reviewer for the suggestion. Since the reviewer did not give any related papers on “Projection networks are common in continual learning”, we have searched for related topics and found some, e.g., [1-3] However, among these related papers, the projection is designed on the parameter/gradient space, i.e., these works are designed to tackle forgetting by finding optimization paths through projection. By contrast, the projection in this paper is designed on the embedding space, which aims to encode task-specific information on pre-trained embeddings. Our method creates and learns parallel subspaces based on the pre-trained vision-language model, while others do not. **Hence, it seems the “projections” are obviously two different concepts.** We are happy to give further clarifications if the reviewer has other related papers for supplementary.
>
> **Furthermore, we need to highlight that the core contribution of our paper is not how to design the projection since they are only linear layers. By contrast, our contributions lie in enabling a pre-trained vision-language model to adapt to downstream tasks continually. Such a task requires the projection expansion strategy, cross-modal fusion, and context joint adaptation. We believe these parts are not common in other papers.**
>
>
> **Q3** The main paper lacks standard deviation in results and doesn't consider multiple runs with different class orders.
>
> **A3** We thank the reviewer for the suggestion. In the main paper, we follow the benchmark class-incremental learning setting [4] to shuffle classes with random seed 1993. Just as the reviewer pointed out, we not only run the paper one time in our experiments. By contrast, **we have compared PROOF to other competitors with five class orders (1993, 1994, 1995, 1996, 1997) and reported the performance curve in Supplementary Section D.1 in the initial submission**. These performance curves contain the average and standard deviation information, and results on CIFAR and ImageNet-R with multiple runs verify that our PROOF consistently outperforms other competitors.

---

> > ### Author Response · Authors · 2023-11-19
> > **Response to Reviewer s6fA (2/3)**
> >
> > **Q4** There's no analysis of time and memory usage, except for a basic mention of memory requirements in supplementary materials.
> >
> > **A4** We thank the reviewer for the suggestion. Running time and memory budget are two important factors for real-world applications. Correspondingly, **we have compared PROOF to other competitors to show the memory budget comparison in Supplementary Section D.2 in the initial submission**. We can find that PROOF uses the same scale parameter as other competitors while having the best performance, showing the memory efficiency of our proposed method.
> >
> > Additionally, according to the reviewer’s comments, we have added the running time (seconds) comparison of different methods on a single Tesla V100 GPU in **Supplementary Section D.3** and report the results here.
> >
> > |Method| CIFAR100 B0 Inc10| Food B0 Inc10|
> > |:----|:----:|:----:|
> > |Finetune |7433|12997|
> > |DualPrompt |	6230|10370|
> > | L2P |	6742 |10752|
> > | iCaRL|	10678| 17109|
> > | MEMO | 7046|	11853|
> > |DER|9102|	15825|
> > |PROOF| **4518**| **8697**|
> >
> > As we can infer from the table, PROOF has the lowest running time compared to other competitors. It indicates that PROOF has the potential to run efficiently in real-world applications.
> >
> > **In summary, we show PROOF is efficient in terms of the memory budget or running time.**
> >
> >
> > **Q5** The paper's narration could also be improved.
> >
> > **A5** We thank the reviewer for the suggestion. In this revision, we have **made Section 3 much shorter** by deleting discussions of baseline methods and **increased the length of Section 4 (especially Section 4.2)** to fully introduce the cross-modal fusion module. Additionally, we have **added the requested experimental results and discussions in Supplementary Section D.3, E.4, E.5.** We hope the revision can improve the quality of the narration.
> >
> > **Q6** It looks like the supplementary materials are more informative and present more interesting results w.r.t. the main paper. Why did the authors exclude them from the main paper?
> >
> > **A6** We thank the reviewer for the suggestion. In the main paper, we use class-incremental learning as the most widely applicable continual learning setting and conduct experiments on various class-incremental learning datasets and protocols (main paper Section 5.2, Table 1, Figure 2). We also investigate each component’s influence with ablation studies (main paper Section 5.3, Figure 3). Finally, we also verify PROOF’s potential in other continual learning settings (e.g., continual cross-modal retrieval) and other applications (preserving zero-shot ability) in the main paper Section 5.4 and Figure 4. In the supplementary, we mainly supply additional results and implementation details that cannot be included in the main paper due to the page limit.
> >
> > **However, all essential parts of the supplementary material are mentioned and discussed in the main paper.** For example, Supplementary Section B indicates PROOF can work competitively in continual cross-modal retrieval, which is discussed in the main paper Section 5.4 and Figure 4 (a). Supplementary Section D.10 discusses the results on a new TV series dataset, which is discussed in the main paper Section 5.4 and Figure 4 (b). Supplementary Section C discusses how to maintain the model’s zero-shot performance, and we have also included it in the main paper Section 5.4 and Figure 4 (c). Apart from these parts, other results in the supplementary act as the additional results of the main paper, e.g., the full results or detailed performance in Table 4. **Hence, there is no critical information excluded from the main paper.**

---

> > > ### Author Response · Authors · 2023-11-19
> > > **Response to Reviewer s6fA (3/3)**
> > >
> > > **Q7** The definition of W is not reported in the paper. How W is defined in the context?
> > >
> > > **A7** **The definition of W is clearly defined in the main paper, Section 4.2, Paragraph “How to define the context?”.** Specifically, W is the concatenation of all textual features in the label space, i.e., $W=[P_t(w_1), P_t(w_2),\dots, P_t(w_{|\mathcal{Y}_b|})]$. Among them, $P_t$ is the textual projection function (as defined in Eq. 4), and $w_i$ stands for CLIP’s text embedding of templated input “a photo of a [CLASS]” (as defined below Eq. 2).
> > >
> > > **Q8** Can the authors provide an analysis of the accuracies of the model varying the number of exemplars?
> > >
> > > **A8** We thank the reviewer for the suggestion. In this revision, we have supplied the analysis of the accuracies of the model varying the number of exemplars in **Supplementary Section E.5 and Figure 20**. Specifically, in the main paper, we mainly follow the benchmark setting [4] to utilize 2000 exemplars for rehearsal. To investigate the influence of exemplar number, we vary the exemplar size among [500,1000,1500,2000] and report the results of PROOF, MEMO, and CoOp on CIFAR100 B0 Inc10 setting. We report the results here and refer the reviewer to Supplementary Section E.5 for more details.
> > >
> > > |Exemplar Number |500|1000|1500|2000|
> > > |:----|:----:|:----:|:----:|:----:|
> > > |CoOp|64.07|69.78|72.53|73.36|
> > > |MEMO|71.13|73.50|74.71|74.98|
> > > |PROOF|**75.50**|**77.58**|**78.45**|**79.05**|
> > >
> > > As we can infer from the results, there are two main conclusions. 1) All methods benefit from more exemplars, and they achieve better performance as the exemplar increases. 2) However, PROOF consistently achieves the best performance among all methods at different exemplar scales. Experiments verify the effectiveness of PROOF in various settings.
> > >
> > >
> > > [1] Gradient projection memory for continual learning. ICLR 2021
> > >
> > > [2] Flattening sharpness for dynamic gradient projection memory benefits continual learning. NeurIPS 2021
> > >
> > > [3] TRGP: Trust Region Gradient Projection for Continual Learning. ICLR 2022
> > >
> > > [4] iCaRL: Incremental Classifier and Representation Learning. CVPR 2017

---

> ### Author Response · Authors · 2023-11-21
> **Looking Forward to Further Discussions**
>
> Dear Reviwer s6fA,
>
> We sincerely appreciate your great efforts in reviewing this paper. Your constructive advice and valuable comments really help improve our paper. Considering the approaching deadline, please let us know if you have follow-up concerns. We sincerely hope you can consider our reply in your assessment, and we can further address unclear explanations and remaining concerns, if any.
>
> Once more, we appreciate the time and effort you've dedicated to our paper.
>
> Best regards,
>
> Authors

---

> ### Comment · Reviewer_s6fA · 2023-11-22
>
> I thank the authors for the answers, however, I have still some concerns about the innovation and applicative scenarios.

---

> > ### Author Response · Authors · 2023-11-22
> >
> > We sincerely appreciate your great efforts in reviewing and joining the discussions of this paper.
> >
> > We first summarize the prior discussions. In our rebuttal, we have tried our best to supply experiments and revise the manuscript. As the reviewer comments, only concerns about the innovation and applicative scenarios are not clearly addressed. **Hence, we thank the reviewer for being satisfied with our efforts on different class orders (Q3), memory usage and running time (Q4), narration (Q5), paper organization (Q6), definitions (Q7), and analysis of exemplars (Q8).**
> >
> > Afterward, we would like to further discuss with the reviewer on **applicative scenarios.** As the reviewer commented in the initial review, "**It would be more interesting to see its performance on tasks like incremental captioning or retrieval.**" Correspondingly, in this paper, we design learning mechanisms to enable a pre-trained vision-language model to learn new tasks continually. To achieve this goal, we first design additive projections based on the frozen encoder so that pre-trained features can be optimized to trace task-specific features. Additionally, to promote cross-modal fusion and contextualized classification, we design a cross-modal fusion module for better-contextualized embeddings. We also utilize visual and textual prototypes as the context information and design a novel context prompt to serve as task anchors to resist forgetting. **We present PROOF as a universal algorithm that can be applied to various continual learning scenarios and various vision-language models. Apart from the benchmark class-incremental learning setting against other state-of-the-art methods, we also conduct experiments on continual cross-modal retrieval tasks with other vision language models (BEiT-3) in the main paper Section 5.4**, showing PROOF still outperforms other competitors by a substantial margin. Such experiments are already contained in the initial submission, which adequately fits the reviewer's comments that "it would be more interesting to see its performance on tasks like incremental captioning or retrieval."
> >
> > **Hence, the reviewer's comments make us confused on this aspect. On the one hand, the reviewer is expecting "its performance on tasks like incremental captioning or retrieval." On the other hand, we have shown that the initial submission already contained the required results, but the reviewer still says, "having concerns on applicative scenarios."**
> >
> > After tasking about the applicative scenarios, we move on to the "**innovation**" part. As the reviewer claimed, "the PROOF mechanism, while innovative, lacks depth. Projection networks are common in continual learning, and the new context definition isn't explored." **Although the reviewer did not mention any related papers supporting "Projection networks are common in continual learning," we have shown in the previous rebuttal that related papers on projections and continual learning are working on the gradient aspect. These works aim to project gradient updating directions not to harm existing knowledge. By contrast, our paper works on feature projection, which projects the pre-trained features into parallel subspaces to build task-specific features. Hence, they differ substantially from the motivation to the operation, except for the similar name of "projection."** As we commented in the prior rebuttal, we sincerely hope the reviewer can provide any other related papers to support the claim of "having concerns on the innovation," we will try our best to tell the difference.
> >
> > Finally, we highlight again our contributions in this paper. Through expandable task-specific projections, we enable a pre-trained vision-language model for continual learning. We also designed a cross-modal fusion module to contextualize the embeddings and less forgetting. **The ideas of learning task-specific projections, learning cross-modal fusion with context information, and joint adaptation with context prompts are innovative and have not been explored by any prior work. PROOF is a unified framework that can be applied to various continual learning scenarios and various vision-language models, showing strong performance in various applicative scenarios.**
> >
> > **Once more, we appreciate the time and effort you've dedicated to our paper. We sincerely hope the reviewer can review our rebuttal to dispel the misunderstanding on the innovation and applicative scenarios.**

---

> > ### Author Response · Authors · 2023-11-23
> > **Thank you for raising the score!**
> >
> > We sincerely appreciate the time and effort you have invested in reviewing our paper, and your insightful feedback has been a critical factor in enhancing the overall quality of our work. Following your latest comments, **we have revised the paper to highlight our innovation and application scenarios**. Specifically,
> > - In **Abstract, Section 1 and Section 3.1**, we highlight the diverse application scenarios of PROOF, including class-incremental learning and continual cross-modal retrieval.
> > - In **Section 2**, we discuss related works on gradient projections and show the differences between PROOF and these works.
> > - In **Section 5.4**, we add more details and experimental results about the continual cross-modal retrieval task in order to highlight the various application scenarios of PROOF.
> > - In **Supplementary Section E.6**, we fully discuss the differences between our work and related topics in gradient projection. We show that these works differ from idea to practice and highlight our main contributions in enabling a vision-language model to continual learning tasks.
> >
> > We have highlighted the revised part in our manuscript in **blue** color. Following your suggestions, we have made the innovation for our paper much clearer, added more discussions about related works, added more details and results of other continual learning scenarios, and tried our best to address all other concerns.
> >
> > As the discussion period is approaching its conclusion, we are eagerly anticipating your thoughts on our revision. Your additional feedback at this stage would be invaluable to us.
> >
> > Please let us know if there are any further clarifications or specific experiments you would recommend to strengthen our paper. We are committed to engaging in a productive dialogue to address any residual issues.
> >
> > Moreover, if our revisions and detailed responses meet the concerns initially raised, we would be grateful if you could consider supporting the paper by increasing the score. Your acknowledgment of our efforts in improving the paper would be immensely appreciated.
> > Thank you once again for your invaluable contributions to refining our research.
> >
> > Best regards,
> >
> > Authors

---

> > > ### Comment · Reviewer_s6fA · 2023-11-23
> > >
> > > I carefully revised the new version of the paper and raised the score to 5.

---

> > > > ### Author Response · Authors · 2023-11-23
> > > > **Thank you for recognizing our response!**
> > > >
> > > > We sincerely appreciate your great efforts in reviewing this paper. In the previous discussions, the reviewer is still concerned about the innovation and applicative scenarios. **Correspondingly, we have shown that the proposed method's innovation does not overlap with existing works, and the proposed method has various application scenarios, including class-incremental learning and continual cross-modal retrieval.**
> > > >
> > > > We would still wonder if there are any further clarifications or specific experiments you would recommend to strengthen our paper in the last several hours. For example, suppose innovation is a core factor leading to a negative comment. In that case, we would like to know if the reviewer can list some related works supporting such a claim, and we will try our best to clarify. Similarly, suppose the application scenario leads to a negative comment. In that case, we also wonder if the reviewer has any questions about the experiments about continual cross-modal retrieval (suggested by the reviewer in the initial comment). We will try our best to address your remaining concerns, if any.
> > > >
> > > > Once more, we appreciate the time and effort you've dedicated to our paper.
> > > >
> > > > Best regards,
> > > >
> > > > Authors

---

### Official Review · Reviewer_VzW8 · 2023-10-31

**Soundness:** 2 fair
**Presentation:** 2 fair
**Contribution:** 2 fair
**Rating:** 6
**Confidence:** 5

**Summary:**

Prior works only focus on the visual branch of CLIP for incremental learning. This paper argues both modalities are important.
- PROOF freezes the image and text encoders of the pre-trained VLM (e.g. CLIP). These contain the generalizable representations learned during pre-training.
- For each new incremental task, it adds new projection layers (P_i, P_t) on top of the frozen encoders. These projections are task-specific.
- When a new task arrives, only the parameters of the projections for the new task are trained. The old projections remain frozen.
- Cross-modal attention fusion is used to adjust the query embedding using context like prototypes and prompts. This allows utilizing both visual and textual information to obtain comprehensive embeddings.
- At inference time, the projections are aggregated to obtain a unified classification. But the old projections remain unchanged.

**Strengths:**

Technically a novel idea to incorporate both the visual and the text encoders.
Improves upon SOTA.

**Weaknesses:**

- Inference Mismatch - Projections are combined at inference time which may not fully match the training conditions for a specific task projection.

- Representation Drift - The post-attention module representations learned by the frozen projections may drift or shift slightly during new task training due to weight updates elsewhere. Small drifts can accumulate.

- Section 3 is really long and has a lot of redundant information, it should be made much shorter. That space should be given to increase the length of section 4 to give a better understanding of the fusion module.

**Questions:**

- Any comments on the issues pointed out in the weaknesses will be appreciated.

- Also please make it more clear how you are using attention.

---

> ### Author Response · Authors · 2023-11-19
> **Response to Reviewer VzW8 (1/3)**
>
> We thank the reviewer for the feedback and comments. We have carefully revised the manuscript following the reviewer’s comments, and the equation/section numbers in the rebuttal refer to the latest version. We respond to the concerns below:
>
> **Q1** Inference Mismatch - Projections are combined at inference time which may not fully match the training conditions for a specific task projection.
>
> **A1** We thank the reviewer for the question. We first review the training process of PROOF to illustrate why inference mismatch will not happen. Given the pre-trained vision-language model, we freeze the pre-trained weights and learn expandable projections to map image/text embeddings into task-specific space. Since we learn expandable projections as data evolves, we utilize the additive result as the final image/text embedding (i.e., $P_i(z)$ and $P_t(w)$ in Eq. 4). During training, we freeze former tasks’ projections and only train the current one. The training target is to match the additive features of image-text pair $P_i(z)$ and $P_t(w)$, and the latest projection (the current task’s projection) is optimized to serve as **residual terms** that can adjust the embedding for a holistic prediction. **Since the loss term in Eq. 7 is optimized with exemplars (i.e., a few old class instances) and the current dataset, optimizing it will make the aggregated features holistic to consider all classes.** In other words, since the target is to learn new classes and meanwhile maintain the performance of old classes, being ‘ideal’ projections for some specific task is not the best solution for all seen classes, and our learning protocol strikes a balance between learning new and remembering old. Even if additional projection terms are added after some specific tasks, optimizing Eq. 7 with exemplars and the current dataset still leads to a well-calibrated prediction considering all seen classes.
>
> To empirically show the forgetting degree of different tasks, we also measure the relative forgetting degree using the “Forgetting Measure” in [1][2]. Considering each task’s forgetting degree, it measures the gap between the best and final performance and averages among all tasks. Hence, methods with a lower forgetting measure show better performance in resisting catastrophic forgetting. In this revision, we have supplied the results in **Supplementary Section E.4** and report the results here. We report the results compared to three competitors, i.e., CoOp, iCaRL, and MEMO.
>
>
> | Method | Exemplar | Aircraft B0 Inc10| Cars B0 Inc10| CIFAR100 B0 Inc10| UCF B0 Inc10|
> |:----|:----:|:----:|:----:|:----:|:----:|
> |CoOp| &#10003;| 28.20|	7.43	|	20.74|	10.81|
> |iCaRL | &#10003;| 13.44|	4.18	|	31.45|	6.66|
> |MEMO| &#10003;| 17.34|	4.77|		12.40|	9.78|
> |PROOF| &#10003;| **9.41**| **3.51**| **8.02**| **4.07**|
>
> As shown in the table, **PROOF has the lowest forgetting measure, indicating it suffers the least forgetting (less than 10%) compared to other competitors.** It verifies that the final performance and best performance are almost the same even after long training stages for PROOF. In other words,
> the optimization target makes the latest projection a residual term to **learn a unified embedding considering all seen classes, and inference mismatch can be alleviated**.

---

> > ### Author Response · Authors · 2023-11-19
> > **Response to Reviewer VzW8 (2/3)**
> >
> > **Q2** Representation Drift - The post-attention module representations learned by the frozen projections may drift or shift slightly during new task training due to weight updates elsewhere. Small drifts can accumulate.
> >
> > **A2** We thank the reviewer for the suggestion. It must be noted that there are two main steps in our method, i.e., feature extraction and cross-modal fusion. The extraction process contains the pre-trained embedding and projection, which is designed to extract task-specific features. By contrast, the attention module is responsible for fusing cross-modal information so that features can be more discriminative. **Hence, such fusion is irrelevant to the task, which should be generalizable across tasks. In other words, even if the features shift slightly, the attention process still works to fuse and highlight task-specific information.**
> >
> > On the other hand, the whole optimization flow is continuous, even if the feature representation is changing due to projections. In other words, the drift of former extracted features will result in the corresponding adaptation of the attention module. Since the model is trained with exemplars and current datasets (as defined in Eq. 7), optimizing the loss considers all essential features among all seen classes for decision. The representation drift of former stages will be compensated by the latter stages.
> >
> > Finally, we must highlight an essential module in the attention process, i.e., the context prompt C. It serves as the task anchor for each task, which is specially learned for each new task. Since the cross-modal fusion process will contextualize the visual embedding with the context information, the context prompt can identify the specific task to highlight important features. As a result, the negative impact of representation drift can be alleviated in our method.

---

> > > ### Author Response · Authors · 2023-11-19
> > > **Response to Reviewer VzW8 (3/3)**
> > >
> > > **Q3/Q5** Section 3 is really long and has a lot of redundant information, it should be made much shorter. That space should be given to increase the length of section 4 to give a better understanding of the fusion module. Also please make it more clear how you are using attention.
> > >
> > > **A3** We thank the reviewer for the suggestion. In this revision, we have **made Section 3 much shorter** by deleting discussions of baseline methods and **increased the length of Section 4 (especially Section 4.2)** to fully introduce the cross-modal fusion module. In the following part, we elaborate on the attention mechanism for cross-modal fusion.
> > >
> > > The typical matching target of CLIP is to match the visual feature to the textual feature and output the target with the largest similarity. However, we argue that it would be beneficial to further refine these features to capture the **contextual relationship** between images and texts. For example, when the query instance is a “panda,” it is desirable to adjust the visual features to highlight the discriminative attributes such as **black eyes and ears**. Meanwhile, considering the visual embedding of a panda, the textual features should also be adapted in a coherent manner so that co-adapted visual and textual features can lead to more discriminative predictions. Similarly, when the query instance is a “cat,” features like beards and tails should be emphasized jointly for visual and textual embeddings. This adjustment process involves jointly adapting the query embedding and the context (e.g., textual information) to obtain a contextualized embedding. A desirable adjustment function should be able to relate every other component as context to conduct joint adaptation.  Correspondingly, we use the self-attention mechanism that contextualizes and fuses the query embeddings and cross-modal information.
> > >
> > > We define the context information as $Context=[P, W]$ (we do not consider the context prompt in this part), including the visual prototypes and textual classifiers. Hence, we feed the self-attention module with $[P_i(z), Context]$ as its query ($\mathcal{Q}$), key ($\mathcal{K}$), value ($\mathcal{V}$). During the attention process, it calculates the local similarity among these inputs, and adjusts them simultaneously via the relationship to other components. During the calculation, it first projects $\mathcal{Q}$, $\mathcal{K}$, $\mathcal{V}$ into the same embedding space and calculates the relative similarity. For the visual feature $P_i(z)$, the output is denoted as $P_i(z)+\sum_k\alpha_{qk}V_{:,k}$, i.e., adding a residual term to the original input. The residual information is the weighted combination of other inputs, which corresponds to the visual prototypes and textual classifiers. The adaption is the same for other components in the attention module, and the cross-modal fusion leads to more discriminative features (as shown in the main paper, Figure 3 (b)).
> > >
> > >
> > > **Q4** Any comments on the issues pointed out in the weaknesses will be appreciated.
> > >
> > > **A4** We thank the reviewer for the suggestion. We have taken careful consideration of the above issues, made detailed responses and revised our paper. We sincerely acknowledge the reviewer’s comments that help us improve the quality of this paper and are happy to address other remaining concerns during the following discussion.
> > >
> > > [1] A Comprehensive Survey of Continual Learning: Theory, Method and Application. arXiv 2023.
> > >
> > > [2] Online continual learning in image classification: An empirical survey. Neurocomputing 2022.

---

> ### Author Response · Authors · 2023-11-21
> **Looking Forward to Further Discussions**
>
> Dear Reviwer VzW8,
>
> We sincerely appreciate your great efforts in reviewing this paper. Your constructive advice and valuable comments really help improve our paper. Considering the approaching deadline, please let us know if you have follow-up concerns. We sincerely hope you can consider our reply in your assessment, and we can further address unclear explanations and remaining concerns, if any.
>
> Once more, we appreciate the time and effort you've dedicated to our paper.
>
> Best regards,
>
> Authors

---

> > ### Comment · Reviewer_VzW8 · 2023-11-22
> >
> > Thanks to the Authors for the extended explanations.
> >
> > They are certainly helpful. Although they improve my view of the paper, I am confirming my initial rating.

---

> > > ### Author Response · Authors · 2023-11-22
> > > **Thank you for recognizing our response!**
> > >
> > > Thank you for the time and effort you've dedicated to our paper and for being positive. We sincerely appreciate the time and effort you've dedicated to our paper.

---

### Official Review · Reviewer_asYX · 2023-11-01

**Soundness:** 3 good
**Presentation:** 3 good
**Contribution:** 2 fair
**Rating:** 6
**Confidence:** 5

**Summary:**

This paper proposes a class-incremental learning (CIL) method based on vision-language models. Specifically, this paper mainly focuses on two key challenges to CIL, named how to adapt the model without forgetting and how to make full use of the multi-modal information. To deal with the first challenge, a task-specific projections are proposed based on the frozen image/text encoders. To deal with the second challenge, a fusion module is proposed for better exploit the cross-modality information. Experiments have shown the state-of-the-art performance of the proposed method.

**Strengths:**

- In general, the proposed method is well motivated and clearly presented.
- The paper turns a VLM into a continual learner that is both retentive and comprehensive.
- Good performance is achieved.

**Weaknesses:**

- The effectiveness of alleviating forgetting is uncertain. The process involves incrementally learning image projection heads and text projection heads, which are then combined for various tasks. When new tasks are learned, the projections of previous tasks are fixed and not updated. However, during inference, the projections of all tasks are merged, which might not be ideal for test data from older tasks due to potential side effects caused by the projections from the new tasks.
- The extent to which contextual information is effective has not been extensively studied. The projection fusion method proposes to contextualize and merge embeddings and contextual information using self-attention. However, in the experiments, only the results of Projection & Fusion are compared with Projection & Fusion & Context Prompt, without explicitly evaluating the effectiveness of the concatenated context information in Q, K, V as [P_i(z), Context] in self-attention, or the effectiveness of the context prompt. In other words, the final context information is defined as Context = [P, W, C], but the specific contributions of W and C to the final results need further analysis.
- The evaluation metric used may not provide a comprehensive measure of the extent of forgetting.

**Questions:**

- To what extent the proposed method could alleviate forgetting?
- How does each component of the contextual information contribute to the final results?

---

> ### Author Response · Authors · 2023-11-19
> **Response to Reviewer asYX (1/2)**
>
> We thank the reviewer for the feedback and comments. We have carefully revised the manuscript following the reviewer’s comments, and the equation/section numbers in the rebuttal refer to the latest version. We respond to the concerns below:
>
> **Q1** The effectiveness of alleviating forgetting is uncertain. The process involves incrementally learning image projection heads and text projection heads, which are then combined for various tasks. When new tasks are learned, the projections of previous tasks are fixed and not updated. However, during inference, the projections of all tasks are merged, which might not be ideal for test data from older tasks due to potential side effects caused by the projections from the new tasks.
>
> **A1** We thank the reviewer for the question and answer it from two aspects. We first review the training process of PROOF to illustrate its rationale for overcoming forgetting. Given the pre-trained vision-language model, we freeze the pre-trained weights and learn expandable projections to map image/text embeddings into task-specific space. Since we learn expandable projections as data evolves, we utilize the additive result as the final image/text embedding (i.e., $P_i(z)$ and $P_t(w)$ in Eq. 4). During training, we freeze former tasks’ projections and only train the current one. The training target is to match the additive features of image-text pair $P_i(z)$ and $P_t(w)$, and the latest projection (the current task’s projection) is optimized to serve as **residual terms** that can adjust the embedding for a holistic prediction. **Since the loss term in Eq. 7 is optimized with exemplars (i.e., a few old class instances) and the current dataset, optimizing it will make the aggregated features holistic to consider all classes.** In other words, since the target is to learn new classes and meanwhile maintain the performance of old classes, being ‘ideal’ projections for old classes is not the best solution for all seen classes, and our learning protocol strikes a balance between learning new and remembering old.
>
> Secondly, we also measure the relative forgetting degree using the “Forgetting Measure” in [1][2]. Considering each task’s forgetting degree, it measures the gap between the best and final performance and averages among all tasks. Hence, methods with a lower forgetting measure show better performance in resisting catastrophic forgetting. In this revision, we have supplied the results in **Supplementary Section E.4** and report the results here. We report the results compared to three competitors, i.e., CoOp, iCaRL, and MEMO.
>
>
> | Method | Exemplar | Aircraft B0 Inc10| Cars B0 Inc10| CIFAR100 B0 Inc10| UCF B0 Inc10|
> |:----|:----:|:----:|:----:|:----:|:----:|
> |CoOp| &#10003;| 28.20|	7.43	|	20.74|	10.81|
> |iCaRL | &#10003;| 13.44|	4.18	|	31.45|	6.66|
> |MEMO| &#10003;| 17.34|	4.77|		12.40|	9.78|
> |PROOF| &#10003;| **9.41**| **3.51**| **8.02**| **4.07**|
>
> As shown in the table, **PROOF has the lowest forgetting measure, indicating it suffers the least forgetting (less than 10%) compared to other competitors.** It verifies that the final performance and best performance are almost the same even after long training stages for PROOF.
>
> In summary, the optimization target makes the latest projection a residual term to **learn a unified embedding considering all seen classes**. Besides, experimental results also verify that PROOF shows the lowest forgetting among all competitors.

---

> > ### Author Response · Authors · 2023-11-19
> > **Response to Reviewer asYX (2/2)**
> >
> > **Q2/Q5** The extent to which contextual information is effective has not been extensively studied. The projection fusion method proposes to contextualize and merge embeddings and contextual information using self-attention. However, in the experiments, only the results of Projection & Fusion are compared with Projection & Fusion & Context Prompt, without explicitly evaluating the effectiveness of the concatenated context information in Q, K, V as [P_i(z), Context] in self-attention, or the effectiveness of the context prompt. In other words, the final context information is defined as Context = [P, W, C], but the specific contributions of W and C to the final results need further analysis. How does each component of the contextual information contribute to the final results?
> >
> > **A2** We thank the reviewer for the suggestion. In fact, the ablation study on the context components is extensively researched in **Supplementary Section D.5** in the submitted version. We explain the general results here and refer the reviewer to supplementary materials for more details. Specifically, we conduct experiments by varying the compositional components in the Context, i.e., Context=[P], Context=[W], Context=[P, W], Context=[P, W, C]. For these variants, we keep the other settings the same and report the incremental performance.
> >
> > From the results, we observe that using visual prototypes or textual prototypes alone yields similar performance, and the impact of adjustment is marginal. However, when visual and textual prototypes are jointly utilized as context information, the model can learn from cross-modality and achieve better performance. Lastly, the introduction of context prompts into the context further enhances the performance of PROOF, resulting in the best performance among all variations. These ablation studies verify that each component in the context helps in the final prediction.
> >
> >
> > **Q3/Q4** The evaluation metric used may not provide a comprehensive measure of the extent of forgetting. To what extent the proposed method could alleviate forgetting?
> >
> > **A3** We thank the reviewer for the suggestion. As we responded in A1, we consider the forgetting measure in [1][2] to measure the task-wise forgetting degree. Results show that our proposed PROOF has the lowest forgetting among other competitors. We have included these additional results in **Supplementary Section E.4**.
> >
> > [1] A Comprehensive Survey of Continual Learning: Theory, Method and Application. arXiv 2023.
> >
> > [2] Online continual learning in image classification: An empirical survey. Neurocomputing 2022.

---

> ### Author Response · Authors · 2023-11-21
> **Looking Forward to Further Discussions**
>
> Dear Reviwer asYX,
>
> We sincerely appreciate your great efforts in reviewing this paper. Your constructive advice and valuable comments really help improve our paper. Considering the approaching deadline, please let us know if you have follow-up concerns. We sincerely hope you can consider our reply in your assessment, and we can further address unclear explanations and remaining concerns, if any.
>
> Once more, we appreciate the time and effort you've dedicated to our paper.
>
> Best regards,
>
> Authors

---

> > ### Comment · Reviewer_asYX · 2023-11-22
> > **Post rebuttal**
> >
> > The authors have addressed most of my concerns. I would like to raise my score to 6.

---

> > > ### Author Response · Authors · 2023-11-22
> > > **Thank you for raising the score!**
> > >
> > > Thank you for your recognition of our response and additional experiments! We sincerely appreciate the time and effort you've dedicated to our paper.

---

### Author Response · Authors · 2023-11-19
**General Response**

We would like to express our deepest gratitude to the reviewers for the meticulous examination of the paper and their insightful and valuable comments. We acknowledge that all the reviewers observed the shining point, saying the proposed method is **well motivated and clearly presented, technically a novel idea, the first time a VLM model for continual learning** (asYX, VzW8, s6fA). They also think our proposed method turns a VLM into a continual learner that is both retentive and comprehensive (asYX) and with very interesting and detailed ablation studies (s6fA). Additionally, all the reviewers acknowledge that extensive experiments validate the state-of-the-art performance of our proposed method (asYX, VzW8, s6fA).

In this rebuttal, we have given careful thought to the reviewers’ suggestions and made the following revisions to our manuscript to answer the questions and concerns:

- In **Section 3**, we simplify the descriptions of related baselines to make this part concise.

- In **Section 4**, we carefully revise this section to highlight the rationale of using cross-modal fusion and add the detailed calculation process and illustrations for a better understanding of the cross-modal fusion module.

- In **Supplementary Section D.3**, we add a running time comparison of different methods, which shows our proposed method has the least running time among all competitors.

- In **Supplementary Section E.4**, we add the experiments on the forgetting measure to evaluate the relative forgetting of different tasks. Results show our proposed method has the least forgetting among all competitors.

- In **Supplementary Section E.5**, we add the experiments on changing exemplar numbers. Results show that our proposed method substantially outperforms other competitors using any scale exemplars.

We have highlighted the revised part in our manuscript in **blue** color. Please check the answers to specific comments.

---

### Author Response · Authors · 2023-11-20
**Looking Forward to Further Discussions**

Dear reviewers,

Hope this message finds you well.

We have updated the manuscript according to your comments and responded in detail to your questions. As the discussion period will end in less than three days, we would like to kindly ask whether there are any additional concerns or questions we might be able to address.

Thanks very much for your effort!

Best regards,

Authors

---

### Meta-Review · Area_Chair_w9RC · 2024-01-03

**Metareview:**

The present paper focuses on class incremental learning, where the goal is to update a pretrained model to add classes that were not seen during earlier training phases. Ideally, such methods should learn new classes with little data and compute, and should retain performance on previously learned classes. The authors propose the use of large vision language models (VLMs, here: CLIP), and utilize task-specific projection and multi-modal fusion.

Initial reviews commended the authors for the interesting idea of turning a VLM into a continual learner, and for the strong empirical performance. At the same time, they noted several weaknesses. These ranged from technical aspects (e.g., lack of analysis of time and memory usage, some results missing standard deviations, inference mismatch between training and inference time, representation drift) as well as concerns about whether the depth of insight and significance of the paper was substantial enough for warranting acceptance.

The authors submitted a comprehensive rebuttal including additional empirical analysis. In addition, reviewer-author discussion took place during the rebuttal, and reviewer-AC discussion after the rebuttal phase are taken into account.

All reviewers agree that the paper was substantially improved during the rebuttal and discussion. Technical questions were clarified and many concerns were addressed. At the same time the paper very much remains borderline. Two reviewers contend that the depth of insight provided in the paper is not sufficient for acceptance. In particular, they note that using a powerful model like CLIP for just classification tasks seems somewhat limited, as a much broader range of tasks could be explored. In addition, the proposed method uses relatively widely used components. Given that the main strength of the paper is the strong empirical performance, the reviewers would have expected a wider or deeper set of insights to achieve a more significant contribution.

The AC concurs with the reviewer assessment and recommends rejecting the paper.

**Justification For Why Not Higher Score:**

The paper is technically correct but its depth / breadth of insight does not warrant acceptance.

**Justification For Why Not Lower Score:**

N/A

---

### Decision · Program_Chairs · 2024-01-16

Reject